# Development of a CRISPR/Cas9n-based tool for metabolic engineering of *Pseudomonas putida* for ferulic acid-to-polyhydroxyalkanoate bioconversion

Yueyue Zhou[1], Lu Lin[2 ✉], Heng Wang[3], Zhichao Zhang[3], Jizhong Zhou[4] & Nianzhi Jiao[5,6]

Ferulic acid is a ubiquitous phenolic compound in lignocellulose, which is recognized for its role in the microbial carbon catabolism and industrial value. However, its recalcitrance and toxicity poses a challenge for ferulic acid-to-bioproducts bioconversion. Here, we develop a genome editing strategy for *Pseudomonas putida* KT2440 using an integrated CRISPR/Cas9n-λ-Red system with *pyrF* as a selection marker, which maintains cell viability and genetic stability, increases mutation efficiency, and simplifies genetic manipulation. Via this method, four functional modules, comprised of nine genes involved in ferulic acid catabolism and polyhydroxyalkanoate biosynthesis, were integrated into the genome, generating the KTc9n20 strain. After metabolic engineering and optimization of C/N ratio, poly-hydroxyalkanoate production was increased to ~270 mg/L, coupled with ~20 mM ferulic acid consumption. This study not only establishes a simple and efficient genome editing strategy, but also offers an encouraging example of how to apply this method to improve microbial aromatic compound bioconversion.

[1] Ocean College, Zhejiang University, Zhoushan, Zhejiang, China. [2] Institute of Marine Science and Technology, Shandong University, Qingdao, Shandong, China. [3] Key Laboratory of Health Risk Factors for Seafood of Zhejiang Province, Zhoushan Municipal Center for Disease Control and Prevention, Zhoushan, Zhejiang, China. [4] Institute for Environmental Genomics, Department of Microbiology and Plant Biology, and School of Civil Engineering and Environmental Sciences, University of Oklahoma, Norman, OK, USA. [5] State Key Laboratory of Marine Environmental Science and College of Ocean and Earth Sciences, Xiamen University, Xiamen 361005, China. [6] Joint Lab for Ocean Research and Education at Shandong University, Xiamen University and Dalhousie University, Qingdao, China. ✉email: linlu2019@sdu.edu.cn

Ferulic acid is a ubiquitous phenolic component of lignocellulose, which is linked to hemicellulose and lignin via ester linkages, and is released to the environment by fungal and bacterial esterase enzymes[1–3]. Moreover, ferulic acid is a intermediate metabolite during lignin biodegradation[4,5]. Considering that lignocellulose is an abundant carbon compound on the earth[6], investigation of ferulic acid catabolism not only provides insights into the global carbon cycle[7], but also contributes to the development in biocatalytic conversion of ferulic acid to valuable bioproducts[8,9]. To date, the biotransformation of ferulic acid to vanillin has been intensively studied, as vanillin is the aldehyde intermediate of the ferulic acid degradation pathway, and a widely used flavoring agent[8,10]. Recently there has been strong interest in the use of bacteria to produce more diverse and valuable products (e.g., polyhydroxyalkanoates (PHAs) and muconic acid) from ferulic acid[9,11,12]. PHAs, which act as microbial carbon storage, are a class of natural biopolymers and can be used as environmentally friendly alternatives to petroleum-based plastics, as well as bioimplant materials for medical and therapeutic applications[13–16]. Moreover, vanillic acid, a intermediate metabolite in a ferulic acid catabolism pathway, has been identified as the non-fatty acid feedstock for medium-chain-length PHA (mcl-PHA) biosynthesis[13,17], indicating the feasibility of ferulic acid-to-PHA bioconversion. Hence, ferulic acid-to-PHA conversion provides insights into the microbial carbon sink and ferulic acid valorization.

The gram-negative bacterium *Pseudomonas putida* has emerged as the candidate with the greatest potential for ferulic acid-to-PHA[9,17]. *P. putida* can catabolize ferulic acid via a coenzyme A-dependent, non-β-oxidative pathway[18]. Moreover, it is a well-known *mcl*-PHA-producing organism[13,19]. However, the conversion efficiency is low, posing a challenge for the application of this organism in ferulic acid-to-PHA[9,19]. First, ferulic acid is highly toxic to the bacterial cell, inhibiting cell growth and metabolism[8]. Second, high *mcl*-PHA yields in *P. putida* are more favored to be produced from fatty acids[20,21]. Employing non-fatty acid feedstock, especially aromatic compounds, greatly limits *mcl*-PHA accumulation[19,22]. Hence, metabolic engineering of *P. putida* strains is required to address these two key issues.

A wide variety of genome-editing tools have been developed for metabolic engineering in *Pseudomonas*, the majority of which are based on homologous recombination[23]. Such methods have very low mutation efficiency and are time consuming[23–25]. Although a selection marker was used to make it is easier to select mutant, it usually leaves a marker in the genome[24], and later phage-derived recombinases (e.g., RecET and λ-Red) were employed to improve mutation efficiency[26,27]. However, the process still leaves scars (e.g., 34 bp LoxP site) in the genome and are cumbersome for plasmid curing[28,29]. Moreover, large DNA fragment (>5 kb) insertion has not yet reported via these methods. Type II Clustered Regularly Interspaced Short Palindromic Repeats (CRISPR)-Cas systems are a prokaryotic adaptive immune defense mechanism[30]. Recently, Cas9-induced recombination methods have attracted great attention, due to their high efficiency, ease of design, short manufacturing time and low cost[31,32]. They are classified into two categories: Cas9 and Cas9 nickase (Cas9n) systems. Although both three-plasmid and two-plasmid CRISPR/Cas9-based systems have recently been developed for *P. putida* KT2440[28,33], there are some characteristics that have not been fully investigated. First, CRISPR/Cas9 can cause cell death after double strand breaks (DSB) in bacteria that lack or have low expression of major NHEJ components[34]. Although recombinase proteins (e.g., SSr and λ-Red)[28,33] and specific designing of sgRNA could improve cell survival[35], the effects are variable for different genes and strains[36,37]. Moreover, the strict requirements of sgRNA designing also constrain target

locus selection, especially for organisms with incomplete genome sequences. Second, CRISPR/Cas9 system is known for its high efficiency[32,38]; however, mutation efficiencies of various targets are highly variable, limiting its application in some targets which exhibit low editing efficiency[28,33]. Third, the reported *P. putida* CRISPR/Cas9 systems utilize multiple plasmids and thus presents challenges for transformation and plasmid stability, which is especially evident in continual rounds of genome editing.

Here, we developed an integrated CRISPR/Cas9n-λ-Red genome-editing strategy (CRP) to overcome the above stated limitations. CRISPR/Cas9n, which is rarely reported in *Pseudomonas*, is used as an alternative strategy to avoid bacterial death[31,37,39]. Furthermore, CRISPR/Cas9n and λ-Red expression cassettes were integrated into the chromosome to simplify plasmid construction and decrease the size of the plasmid, whereas the *pyrF* was employed as both a positive and negative marker to ensure theoretical 100% mutation efficiency of any target genes in *P. putida* KT2440. Via this CRP method, we successfully constructed the KTc9n20 strain by modulating the expression of nine different genes that are divided into four engineered modules, to enhance ferulic acid-to-PHA conversion. Through further optimization of cultivation conditions, the *mcl*-PHA production from consumed ferulic acid substrate (20 mM) was increased to ~270 mg/L, laying the foundation for a ferulic acid bioprocessing platform.

## Results

**Variable mutation efficiency of a CRISPR/Cas9-λ-Red system.** This study first tested the previously established CRISPR/Cas9-λ-Red system in *P. putida* KT2440 due to its high mutation efficiency (Table 1 and Supplementary Table 1)[30]. The *pyrF* gene (PP_1815), encoding orotidine-5′-phosphate decarboxylase (ODCase), was chosen as the target, as deletion of this gene would generate uracil auxotrophic and 5-fluoroorotic acid (5-FOA) resistant phenotypes[39]. The results suggested *pyrF* was knocked-out, validating this CRISPR/Cas9-λ-Red system works in KT2440, consistent with the previous study[28] (Supplementary Notes). However, the calculated mutation efficiency for *pyrF* was 1/7730 (Table 2), far from previously reported high mutation efficiency (70–100%)[28]. We speculated that the low mutation rate for *pyrF* was likely due to variant efficiency among various genes[33]. Moreover, considering that various vectors with different copy numbers were used to express the four components (Cas9, the gRNA cassette, λ-Red system and the homologous repairing arms) between this study and previous work[28], expression dose of the four functional elements was also a factor contributing to the varied mutation rates (Table 2 and Supplementary Table 2). Together, highly variable editing efficiency of CRISPR/Cas9-λ-Red system restricts whole genome editing in bacteria in general, and certain genes of *P. putida* in particular, which have either low repairing ability or poor mutation rate.

**Integrating Cas9n and λ-Red into the KTc9n strain.** To increase editing efficiency, we endeavored to develop a CRISPR/Cas9n system[39,40]. The 140 kDa His-tagged Cas9n protein was expressed as evidenced by SDS-PAGE analysis, while it was absent in the control strain KT$_G$ (Supplementary Fig. 1 and Table 1). Subsequently, the two plasmid CRISPR/Cas9n systems, both with and without the λ-Red system, respectively, were tested in KT2440. Via the CRISPR/Cas9n system, *pyrF* deletion efficiency was comparable to that of CRISPR/Cas9-λ-Red system, indicating that the CRISPR/Cas9n system increased editing efficiency (Table 2). Furthermore, when the λ-Red system was introduced, deletion efficiency was increased to 1/2146, and further raised to

**Table 1 Plasmids and bacterial strains used in this study.**

| Strain or plasmid | Relevant characteristics | Source or reference |
|---|---|---|
| Plasmids | | |
| pPROBE-GT | pVS1/p15a Gm gfp | [65] |
| pBBR1MCS2 | pBBR1 oriV Kan | [66] |
| pCas9 | pVS1/p15a Gm, $P_{min}$::cas9, gfp | This study |
| pCas9n | pVS1/p15a Gm, $P_{min}$:: cas9n, gfp | This study |
| pRed | pVS1/p15a Gm, PxylA:: λ-Red cassette, gfp | This study |
| pCas9-Red | pVS1/p15a Gm, $P_{xylA}$:: λ-Red cassette, $P_{min}$:: cas9n, gfp | This study |
| pCas9n1 | pVS1/p15a Gm, Ptrc:: gRNA(pyrF), 0.5 kb pyrF homologous arms, $P_{min}$:: cas9n, gfp | This study |
| pCas9n-Red | pVS1/p15a Gm, $P_{xylA}$:: λ-Red cassette, $P_{min}$:: cas9n, gfp | This study |
| pgRNA | pVS1/p15a Gm, Ptrc:: gRNA(pyrF), gfp | This study |
| pDonor | pVS1/p15a Gm, 0.5 kb pyrF homologous arms, gfp | This study |
| pgRNA-donor | pVS1/p15a Gm, Ptrc:: gRNA(pyrF), 0.5 kb pyrF homologous arms, gfp | This study |
| pBgRNA | pBBR1 oriV Kan, Ptrc:: gRNA(pyrF) | This study |
| pBBR1-1 | pBBR1 oriV Kan, Ptrc:: gRNA(pyrF), 0.5 kb pyrF homologous arms | This study |
| pBBR1-2 | pBBR1 oriV Kan, Ptrc:: gRNA(pyrF), 0.5 kb pyrF donor up-$P_{min}$::cas9n- $P_{xylA}$:: λ-Red cassette- 0.5 kb pyrF donor down | This study |
| pBBR1-3 | pBBR1 oriV Kan, Ptrc:: gRNA(icd), 0.5 kb icd donorup-pyrF-0.5 kb icd donordown, $P_{tac}$::gfp, Pvan:: sgRNAps, palindromic sequence | This study |
| pBBR1-4 | pBBR1 oriV Kan, Ptrc:: gRNA(pyrF), 0.5 kb icd donorup-0.5 kb icd donordown, $P_{tac}$::gfp, Pvan:: sgRNAps, palindromic sequence | This study |
| pBBR1-5 | pBBR1 oriV Kan, Ptrc:: gRNA(tesA), 0.5 kb tesA donorup-pyrF-0.5 kb tesA donordown, $P_{tac}$::gfp, Pvan:: sgRNAps, palindromic sequence | This study |
| pBBR1-6 | pBBR1 oriV Kan, Ptrc:: gRNA(pyrF), 0.5 kb tesA donorup-0.5 kb tesA donordown, $P_{tac}$::gfp, Pvan:: sgRNAps, palindromic sequence | This study |
| pBBR1-7 | pBBR1 oriV Kan, Ptrc:: gRNA(tesB), 0.5 kb tesB donorup-pyrF-0.5 kb tesB donordown, $P_{tac}$::gfp, Pvan:: sgRNAps, palindromic sequence | This study |
| pBBR1-8 | pBBR1 oriV Kan, Ptrc:: gRNA(pyrF), 0.5 kb tesB donorup-0.5 kb tesB donordown, $P_{tac}$::gfp, Pvan:: sgRNAps, palindromic sequence | This study |
| pBBR1-9 | pBBR1 oriV Kan, Ptrc:: gRNA(phaZ), 0.5 kb phaC1 donorup-pyrF-0.5 kb phaC1donordown, $P_{tac}$::gfp, Pvan:: sgRNAps, palindromic sequence | This study |
| pBBR1-10 | pBBR1 oriV Kan, Ptrc:: gRNA(pyrF), 0.5 kb phaC1 donorup-2phac1-0.5 kb phaC1donordown, $P_{tac}$::gfp, Pvan:: sgRNAps, palindromic sequence | This study |
| pBBR1-11 | pBBR1 oriV Kan, Ptrc:: gRNA(pyrF), 0.5 kb TphaC1 donorup-2Tphac1-0.5 kb TphaC1donordown, $P_{tac}$::gfp, Pvan:: sgRNAps, palindromic sequence | This study |
| pBBR1-12 | pBBR1 oriV Kan, Ptrc:: gRNA(fcs), 0.5 kb echvdhfcs donorup-pyrF-0.5 kb echvdhfcs donordown, $P_{tac}$::gfp, Pvan:: sgRNAps, palindromic sequence | This study |
| pBBR1-13 | pBBR1 oriV Kan, Ptrc:: gRNA(pyrF), 0.5 kb echvdhfcs donorup-2echvdhfcs-0.5 kb echvdhfcs donordown, $P_{tac}$::gfp, Pvan:: sgRNAps, palindromic sequence | This study |
| pBBR1-14 | pBBR1 oriV Kan, Ptrc:: gRNA(pyrF), 0.5 kb 3359 donorup-echvdhfcs-0.5 kb echvdhfcs donordown, $P_{tac}$:: gfp, Pvan:: sgRNAps, palindromic sequence | This study |
| pBBR1-15 | pBBR1 oriV Kan, Ptrc:: gRNA(pyrF), 0.5 kb 3359 donorup- echvdhfcs-echvdhfcs-0.5 kb echvdhfcs donordown, $P_{tac}$::gfp, Pvan:: sgRNAps, palindromic sequence | This study |
| pBBR1-16 | pBBR1 oriV Kan, Ptrc:: gRNA(vanAB), 0.5 kb vanAB donorup-pyrF-0.5 kb vanAB donordown, $P_{tac}$::gfp, Pvan:: sgRNAps, palindromic sequence | This study |
| pBBR1-17 | pBBR1 oriV Kan, Ptrc:: gRNA(pyrF), 0.5 kb vanAB donorup-vanAB-vanAB-0.5 kb vanAB donordown, $P_{tac}$:: gfp, Pvan:: sgRNAps, palindromic sequence | This study |
| pBBR1-18 | pBBR1 oriV Kan, Ptrc:: gRNA(pyrF), 0.5 kb vanAB donorup-vanAB-vanAB -vanAB-0.5 kb vanAB donordown, $P_{tac}$::gfp, Pvan:: sgRNAps, palindromic sequence | This study |
| pBBR1-19 | pBBR1 oriV Kan, Ptrc:: gRNA(tesBII), 0.5 kb tesBII donorup-pyrF-0.5 kb tesBII donordown, $P_{tac}$::gfp, Pvan:: sgRNAps, palindromic sequence | This study |
| pBBR1-20 | pBBR1 oriV Kan, Ptrc:: gRNA(pyrF), 0.5 kb tesBII donorup-0.5 kb tesBII donordown, $P_{tac}$::gfp, Pvan:: sgRNAps, palindromic sequence | This study |
| pBBR1-2-1 | pBBR1 oriV Kan, Ptrc:: gRNA(pyrF), 0.5 kb pyrF donor up- pPROBE-GT vector sequence-0.5 kb pyrF donor down | This study |
| Strains | | |
| P. putida KT2440 | Wild type | ATCC |
| $KT_G$ | KT2440 carrying pPROBE-GT | This study |
| KTΔpyrF | KT2440 ΔpyrF | This study |
| KTc9n | KT2440 ΔpyrF, Pmin::cas9n, $P_{xylA}$:: gam-bet-exo | This study |
| KTc9nΔicd | KTc9n Δicd pyrF | This study |
| KTc9nΔvanAB | KTc9n ΔPvan::vanA-vanB pyrF | This study |
| KTc9nΔfcs-ech-vdh | KTc9n Δ$P_{fcs}$:: fcs-ech-vdh pyrF | This study |
| KTpGT | KTc9n pPROBE-GT | This study |
| KTc9n1 | KTc9n $P_{fcs}$::fcs-ech-vdh | This study |
| KTc9n2 | KTc9n1 $P_{fcs}$::fcs-ech-vdh | This study |
| KTc9n3 | KTc9n ΔferR | This study |
| KTc9n4 | KTc9n $P_{fcs}$::fcs-ech-vdh, ΔferR | This study |
| KTc9n5 | KTc9n $P_{van}$::vanA-vanB | This study |
| KTc9n6 | KTc9n $P_{fcs}$:: fcs-ech-vdh $P_{van}$::vanA-vanB | This study |
| KTc9n7 | KTc9n ΔferR, $P_{fcs}$::fcs-ech-vdh, $P_{van}$::vanA-vanB | This study |
| KTc9n8 | KTc9n ΔferR $P_{fcs}$:: fcs-ech-vdh, $P_{van}$::vanA-vanB, $P_{van}$::vanA-vanB | This study |
| KTc9n9 | KTc9n Δicd | This study |
| KTc9n10 | KTc9n ΔtesA | This study |
| KTc9n11 | KTc9n ΔtesB | This study |
| KTc9n12 | KTc9n ΔtesBII | This study |
| KTc9n13 | KTc9n ΔphaZ, $P_{phaC1}$:: phaC1 | This study |
| KTc9n14 | KTc9n $P_{phaC1}$::phaC1(mutant) | This study |
| KTc9n15 | KTc9n ΔtesBΔtesBII | This study |
| KTc9n16 | KTc9n ΔtesBΔtesBII $P_{phaC1}$:: phaC1(mutant) | This study |
| KTc9n17 | KTc9n6 ΔtesBΔtesBII | This study |
| KTc9n18 | KTc9n6ΔtesBΔtesBII $P_{phaC1}$:: phaC1(mutant) | This study |
| KTc9n19 | KTc9n8ΔtesBΔtesBII | This study |
| KTc9n20 | KTc9n8 ΔtesBΔtesBII $P_{phaC1}$::phaC1(mutant) | This study |

**Table 2 Mutation efficiency of different genome-editing methods.**

| Target gene | System | Plasmid | Copy number (copies per cell) | Component | Efficiency |
|---|---|---|---|---|---|
| *pyrF* | Homologous recombination | pBBR1-1 | 60 | 0.5 kb donor template | 1/164990 |
| | λ-Red recombination | pRed | 13 | λ Red recombinase, | 1/42490 |
| | | pBBR1-1 | 60 | donor template | |
| | CRISPR/Cas9 | pCas9 | 15 | Cas9, | 0 |
| | | pBBR1-1 | 60 | gRNA, donor template | |
| | CRISPR/Cas9 and λ-Red recombinase | pCas9-Red | 8 | Cas9, λ Red recombinase, | 1/7730 |
| | | pBBR1-1 | 60 | gRNA, donor template | |
| | CRISPR/Cas9n | pCas9n | 15 | Cas9n, | 1/7259 |
| | | pBBR1-1 | 60 | gRNA, donor template | |
| | CRISPR/Cas9n and λ-Red recombinase | pCas9n-Red | 9 | Cas9n, λ Red recombinase, | 1/10 |
| | | pBBR1-1 | 60 | gRNA, donor template | |

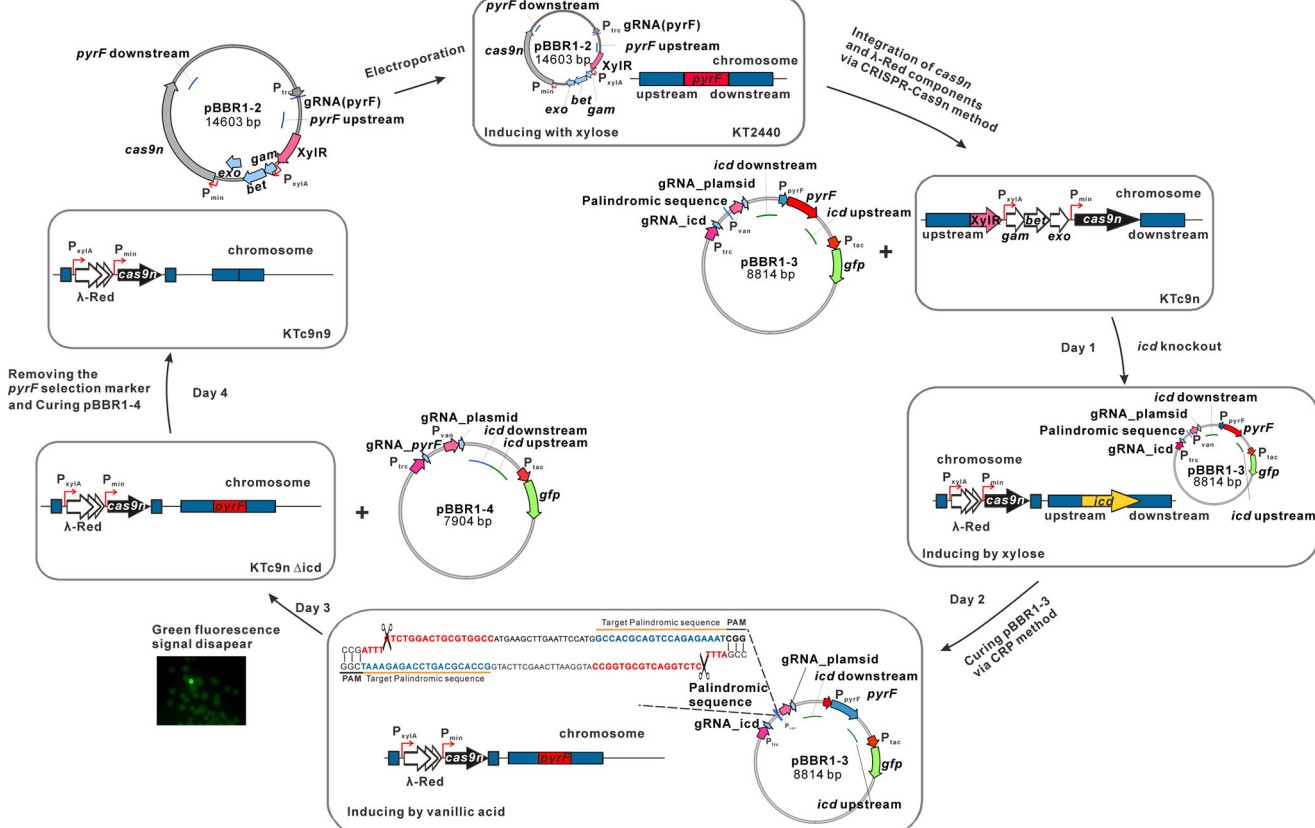

**Fig. 1 Schematic of genome editing in *P. putida* KT2440 via the CRP strategy.** Strain KTc9n was constructed via introduction of the **pBBR1-2** plasmid into *P. putida* KT2440. It integrated the Cas9n and λ-Red expression cassettes into the genome to replace *pyrF*. The procedure for *icd* deletion is shown as an example to illustrate the detailed strategy. Day 1: Electroporation of plasmid **pBBR1-3** into KTc9n for *icd* knockout. Following electroporation, cells were grown on LB pressure medium (xylose+Ura⁺+Km) at 30 °C, 200 rpm for 18 h for genome editing. Day 2: Transfer 1% of cultivated cells into 5 mL fresh 15 mM glucose-M9 minimal selective medium (Ura⁻), supplemented with 4 mM vanillic acid to induce **pBBR1-3** plasmid curing. Colonies without green fluorescence indicated the *icd* gene was deleted and that plasmid **pBBR1-3** was eliminated. Day 3: Introduce **pBBR1-4** plasmid into KTc9nΔ*icd*, to remove the *pyrF* marker. Day 4: Identify KTc9n9 mutant without *pyrF* selection marker from the LB selection plate (Ura⁺+5-FOA).

1/10 through the expression dose optimization of λ-Red recombinase (Table 2). The expression dose was optimized by regulating the promoter activity of $P_{xylA}$ through xylose inducer concentrations (Supplementary Table 2). All of this suggested the CRISPR/Cas9n-λ-Red system was the best method among our tested genome-editing approaches. Next, to decrease the size of the plasmid and simplify the gene editing procedure, $P_{min}$::*cas9n* and $P_{xylA}$:: *gam-bet-exo* expression cassettes were integrated into the *pyrF* gene location on the chromosome of *P. putida* KT2440, to replace *pyrF*, via the CRISPR/Cas9n-λ-Red system (Fig. 1 and Supplementary Table 3). The resulting strain was designated KTc9n (Table 1).

**Precise and high efficiency genome editing by the CRP method.** Although the CRISPR/Cas9n-λ-Red system significantly increased efficiency, it was still far lower than that of other reported CRISPR-Cas9(n) genome-editing systems, which can easily select the mutant without monitoring phenotype[26,28,39]. Therefore, developing a CRISPR/Cas9n-λ-Red system coupled with a selectable marker is essential for those organisms or genes with lower editing efficiency. In this study, the *pyrF* gene was employed as both a positive and negative selection marker, as it is associated with uracil prototrophy and fluoroorotic acid sensitivity[24]. Therefore, KTc9n, which showed no obvious growth difference from KT2440, was used as the host strain for the

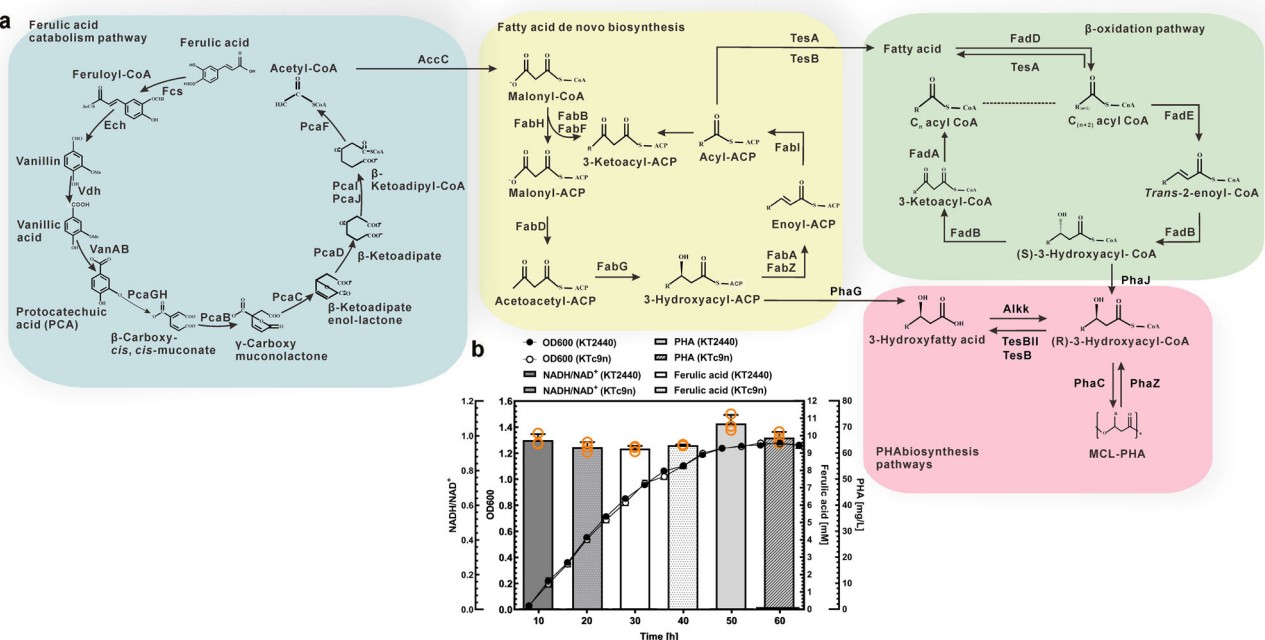

**Fig. 2 Proposed ferulic acid-to-*mcl*-PHA bioconversion route.** **a** Metabolic pathways for ferulic acid-to-*mcl*-PHA bioconversion, including ferulic acid catabolism pathway, fatty acid biosynthesis pathway, β-oxidation pathway and PHA biosynthesis pathway. **b** Cell growth, NADH/NAD$^+$ ratio, residual concentrations of ferulic acid in medium and *mcl*-PHA production by *P. putida* KT2440 and KTc9n. The two strains were grown on M9 mineral medium (65 mg L$^{-1}$ NH$_4$Cl) with 20 mM ferulic acid as the sole carbon substrate ($n = 3$ biologically independent experiments).

following genome editing (Fig. 2 and Supplementary Fig. 2a). The *icd* gene (PP_4011), encoding isocitrate dehydrogenase in the TCA cycle, was chosen as a target gene to test the CRP approach. The **pBBR1-3** plasmid, which contains the P$_{trc}$::sgRNA for *icd*, *pyrF* selection marker and 0.5 kb repair template, was transformed into KTc9n (Fig. 1 and Table 1). And twenty colonies, which were grown on selective plates (M9 + Ura$^-$), were randomly picked for further PCR screening and DNA sequencing. These tests revealed that the deletion rate was 100%, leaving the *pyrF* marker at the target site of the mutant KTc9nΔ*icd* (Fig. 1). Subsequently, the **pBBR1-4** plasmid was transformed into KTc9nΔ*icd*, to remove *pyrF* marker via the same strategy. DNA sequencing for fifteen randomly picked mutants revealed all of the colonies contained a precise scarless deletion of the 1000 bp target sequence of *icd*, generating the mutant KTc9n9 (Supplementary Table 3).

**One-step CRP approach to genome editing and plasmid curing.** Here, we expanded the CRP approach application in plasmid elimination. A palindromic sequence was inserted into the genome-editing plasmids (e.g., **pBBR1-3**) as the target site to ensure Cas9n-induced DSB (Fig. 1). Correspondingly, the gRNA, targeted to the palindromic sequence, was designed and driven by the vanillic acid inducible promoter P$_{van}$[17]. In addition, the reporter gene *gfp* was incorporated into the plasmids to monitor the curing efficiency. Once genome editing was done, 4 mM vanillic acid was added to inducing the suicide function. After 12 h, the loss of signal fluorescence reflected plasmid curing and 90% of the plated colonies were plasmid-free after 24 h (Fig. 1 and Supplementary Fig. 2b). This clearly demonstrated that CRP can be applied to single-step genome editing and plasmid curing.

**Application of CRP genome-editing system in *P. putida* KT2440.** In addition to the *icd* gene, the mutant efficiencies for

other locations in the KT2440 strain were systematically investigated (Supplementary Table 3). The deletion efficiencies were tested for in either single-gene/gene cluster or double-gene with different deleted lengths (from 98 bp to 4643 bp) and were found to be 100% (Supplementary Table 3). Meanwhile, the efficiency for gene integration was also examined. DNA fragments with different lengths (from 1900 bp to 15,000 bp) were successfully integrated into the chromosome of KTc9n, respectively (Supplementary Table 3). Furthermore, the efficiency of simultaneous gene insertion and deletion in the KTc9n genome (KTc9n13) also achieved 100%. For each mutant, 8–20 colonies were randomly picked from the selection plates (LB + 5FOA + Ura$^+$) for PCR and DNA sequencing, and found that all contained the desired mutation (Supplementary Table 3). Hence, our results suggested that the mutation efficiency of this method theoretically can reach 100%, which has an advantage in genome-editing events with low efficiency, such as a large DNA fragment (>5 kb) editing.

**Engineered *P. putida* enhances ferulic acid utilization.** The genome analysis suggested *P. putida* KT2440 employs the coenzyme A-dependent, non-β-oxidative pathway to catabolize ferulic acid. The generated acetyl-CoA enters fatty acid biosynthesis pathway (in addition to replenishing the TCA cycle) and is further channeled to PHA biosynthesis pathways, revealing the feasibility of ferulic acid-to-*mcl*-PHA bioconversion (Fig. 2a). Thus, KT2440 and KTc9n were cultured in 20 mM ferulic acid-M9 mineral medium with 65 mg L$^{-1}$ NH$_4$Cl, respectively. Both strains showed similar cell growth, in which ~10 mM ferulic acid was consumed and ~65 mg L$^{-1}$ *mcl*-PHA was accumulated (Fig. 2b and Supplementary Table 4). No significant difference in ferulic acid utilization ($p = 0.75$), NADH/NAD$^+$ ratio ($p = 0.19$) and *mcl*-PHA production ($p = 0.06$) between KTc9n and KT2440 were observed (Fig. 2b and Supplementary Table 4). These data demonstrated two aspects of these strains. First, the inserted *cas9n* and λ-Red recombinase genes in KT2440 do not influence

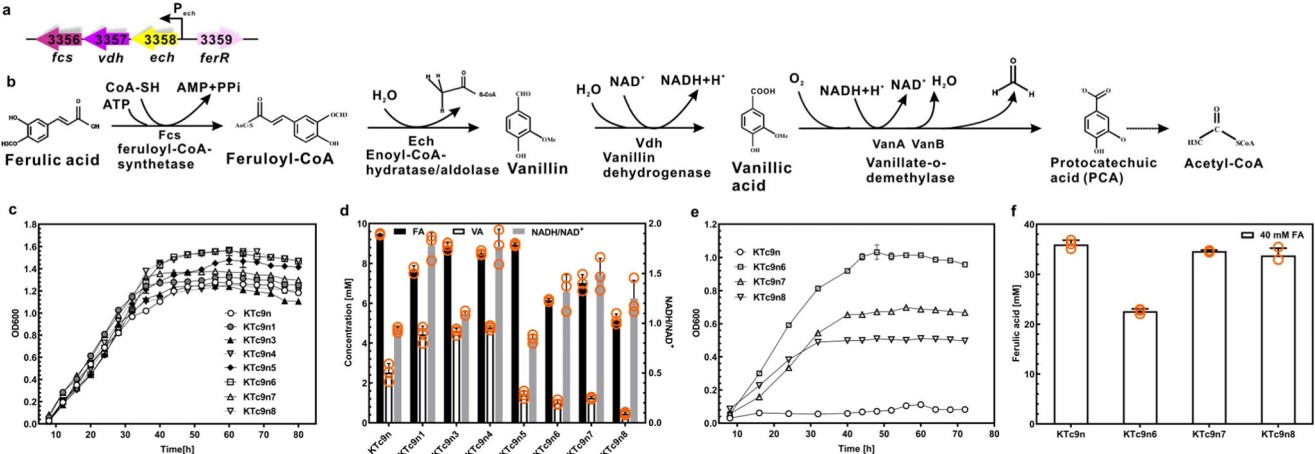

**Fig. 3 Metabolic engineering of *P. putida* KT2440 for enhanced ferulic acid degradation and tolerance. a** Organization of the ferulic acid catabolism locus in KT2440. P: promoter. IDs and gene names of the corresponding enzymes are shown. **b** Ferulic acid catabolism pathway. Dashed arrow indicates multiple steps. **c–d** Cell growth curves, NADH/NAD$^+$ ratio, residual concentrations of ferulic acid and vanillic acid of the mutant strains. They were grown on M9 mineral medium (65 mg L$^{-1}$ NH$_4$Cl) with 20 mM ferulic acid as the sole carbon substrate ($n = 3$ biologically independent experiments). **e, f** Cell growth and residual concentrations of ferulic acid in medium of mutant strains. They were cultured in 40 mM ferulic acid-M9 mineral medium (65 mg L$^{-1}$ NH$_4$Cl) ($n = 3$ biologically independent experiments).

the strain growth and metabolism under such condition. Second, KT2440 and KTc9n have limited capacity for ferulic acid-to-*mcl*-PHA bioconversion. Therefore, KTc9n can be used as an ideal background strain to improve ferulic acid-to-PHA bioconversion.

To improve the ferulic acid utilization performance of KTc9n, the ferulic acid catabolic gene cluster was investigated. The *ech*, *vdh*, and *fcs* genes, which encode feruloyl-CoA hydratase/lyase, vanillin dehydrogenase, and feruloyl-CoA synthetase, respectively, are in one putative operon, while *ferR*, encoding the MarR-type transcriptional regulator, resides in another operon and is located downstream of the *ech-vdh-fcs* gene cluster, with opposite orientation (Fig. 3a). Boosting the expression of *ech-vdh-fcs* gene cluster should be a promising strategy to enhance ferulic acid catabolic capacity (Fig. 3a, b). For this purpose, a second genomic copy of the *ech-vdh-fcs* gene cluster was inserted upstream of *fcs* in the KTc9n, generating the mutant KTc9n1 (Supplementary Table 3). This strain not only showed improved cell growth, but also consumed more ferulic acid (~12 mM, $p = 0.00$), compared to that of KTc9n (~10 mM), validating the hypothesis (Fig. 3c, d). In addition, the MarR-type transcriptional regulator has been reported to negatively regulate the expression of the ferulic acid catabolic operon in *P. fluorescens* BF13 and *Sphingobium* sp. strain SYK-6[41,42], although the role of the homologous protein was not yet well studied in *P. putida* KT2440. Therefore we constructed the KTc9n3 mutant, in which *ferR* was deleted. It exhibited a higher ferulic acid utilization (~11 mM, $p = 0.005$) than that of KTc9n (Fig. 3d).

To further enhance the ferulic acid utilization, the *ech-vdh-fcs* gene cluster was inserted into the genome of KTc9n3 and the resulting mutant was designated KTc9n4 (Table 1). However, neither KTc9n4 cell growth nor ferulic acid consumption were further improved, in comparison to KTc9n1 (Fig. 3c, d). Fcs, Ech and Vdh enzymes convert ferulic acid to vanillic acid, whereas NAD$^+$ is consumed and NADH is generated during this reaction[8] (Fig. 3b). To reveal whether this limitation was caused by intermediate metabolite accumulation and/or redox imbalance, the concentrations of vanillic acid and the ratios of NADH to NAD$^+$ of these mutants were assessed. The concentration of vanillic acid, as generated by KTc9n1, KTc9n3 and KTc9n4, respectively, was higher than that of KTc9n. Of particular note,

KTc9n4 attained the highest level (~4.8 mM, Fig. 3d). Similarly, the NADH/NAD$^+$ ratios of them were higher than that of KTc9n, especially for KTc9n1 and KTc9n4. Together, this indicated KTc9n4 accumulated more vanillic acid and consumed more NAD$^+$ cofactor, resulting in insufficient redox for energy metabolism[8] (Fig. 3b). Previous studies suggested that over-expression of *vanAB* promotes vanillic acid catabolism in *P. putida* KT2440[12] and A514[17], respectively. Therefore, we inferred that increasing the expression of *vanAB*, which regenerates NAD$^+$ from NADH, should counter this hurdle (Fig. 3b).

KTc9n5, carrying a second copy of *vanAB* genes, was constructed (Table 1). Compared to KTc9n, it showed half the vanillic acid concentration and lower NADH/NAD$^+$ ratio. Moreover, more ferulic acid was consumed and the cell growth of KTc9n5 was also improved (Fig. 3c, d). These results suggested *vanAB* plays a role in alleviating vanillic acid accumulation and NAD$^+$ imbalance. Thus, one copy of *vanAB* genes was inserted into the chromosomes of KTc9n1 and KTc9n4, generating the strains KTc9n6 and KTc9n7, respectively (Table 1). As expected, KTc9n6 exhibited greater ferulic acid utilization (~14 mM) and accumulated less vanillic acid (~1 mM) with lower NADH/NAD$^+$ ratio, as compared to KTc9n1 (Fig. 3d). Moreover, its cell growth was greatly improved (Fig. 3c). On the other hand, the performance of KTc9n7 on ferulic acid utilization and cell growth was improved, and NADH/NAD$^+$ ratio decreased, in contrast to KTc9n4 (Fig. 3d). However, KTc9n7 did not outperform the KTc9n6 strain, exhibiting relatively poorer cell growth and ferulic acid utilization (Fig. 3c, d). This might have been the result of an inappropriate gene expression ratio. Unlike KTc9n6, simultaneous single copy *ech-vdh-fcs* gene cluster insertion and *ferR* gene deletion in the genome of KTc9n7 should greatly accelerate the ferulic acid to vanillic acid conversion, whereas a single copy *vanAB* genes insertion might be insufficient for vanillic acid catabolism and cofactor requirement. This speculation was validated when, in contrast to KTc9n6, higher concentration of vanillic acid and NADH/NAD$^+$ ratio were observed in KTc9n7 (Fig. 3d). Therefore, two copies of the *vanAB* genes were inserted into the KTc9n4 genome, constructing the mutant KTc9n8 (Table 1). Consequently, KTc9n8 showed the best performance in cell growth and ferulic acid consumption (~15 mM) among the mutants, which

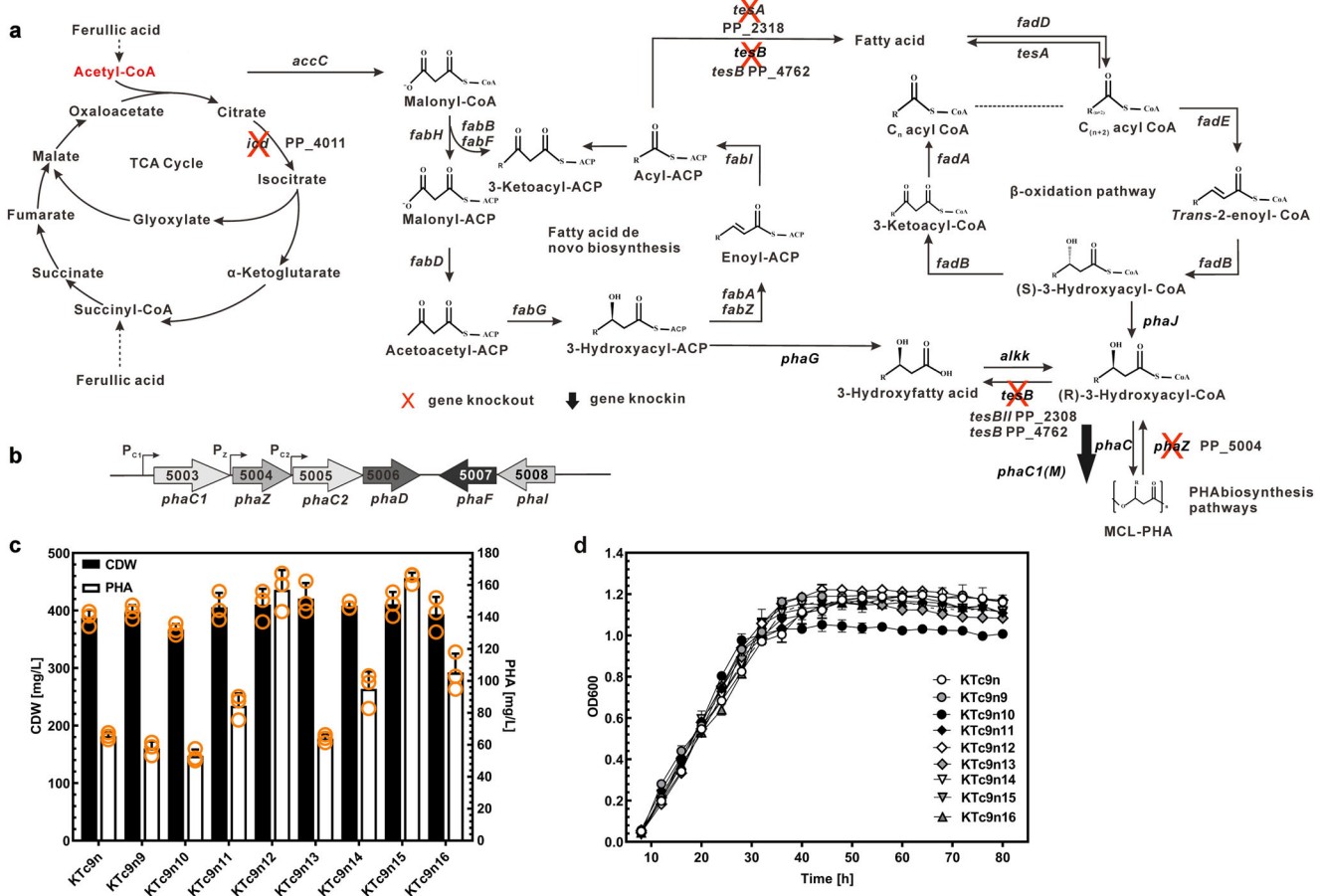

**Fig. 4 Metabolic engineering of *P. putida* KT2440 for enhanced mcl-PHA production. a** Schematic of *mcl*-PHA biosynthesis pathways that are involved in TCA cycle, fatty acid biosynthesis and β-oxidation. Genes that are knocked-out and knocked-in in KTc9n are marked. **b** Organization of the PHA biosynthetic genes of KT2440. IDs and gene names of the corresponding enzymes are shown. **c, d** The cell dry weight, PHA titer and cell growth of strains under 20 mM ferulic acid-65 mg L$^{-1}$ NH$_4$Cl condition (n = 3 biologically independent experiments).

was consistent with its lowest concentration of vanillic acid (~0.46 mM) and lower NADH/NAD$^+$ ratio (Fig. 3c, d).

Finally, the three mutants (KTc9n6, Ktc9n7 and KTc9n8) with higher ferulic acid degradation capacity were further examined for their robustness in the presence of 40 mM ferulic acid. All the strains could grow with 40 mM ferulic acid as the feedstock, while the reference strain, KTc9n, was unable to grow (Fig. 3e). Among them, KTc9n6 showed the greatest robustness, with excellent cell growth and ferulic acid utilization (Fig. 3e, f). Hence, KTc9n6 with the best robustness and KTc9n8 with the greatest ferulic acid utilization would be employed for the remainder of this study.

**Engineered *P. putida* increases *mcl*-PHA biosynthesis.** *mcl*-PHA is produced from fatty acid biosynthesis pathway, when *P. putida* utilizes lignin-derived substrates as carbon sources[43] (Fig. 4a). Previous studies reported that inactivation of isocitrate dehydrogenase (*icd*) increased the flux of acetyl-CoA into fatty acid biosynthesis pathway[44] and was proposed to be a genetic engineering target to improve PHA production. However, *icd* deletion (KTc9n9, Table 1), here, did not contribute to *mcl*-PHA synthesis (Fig. 4c, d, Supplementary Notes and Supplementary Table 4).

Alternatively, the effect of inhibiting the reverse reaction, connecting fatty acid biosynthesis pathway and PHA biosynthesis pathway, was tested (Fig. 4a). Three thioesterase genes *tesA*, *tesB*, and *tesBII*, which catalyze the acyl-ACP or 3-hydroxyacyl-CoA to corresponding acid[45], respectively, were deleted (Fig. 4a). The

resulting mutants were designated KTc9n10, KTc9n11, and KTc9n12 (Table 1 and Supplementary Table 3). Their cell growth and *mcl*-PHA titers were assessed, while KTc9n was the control. KTc9n10 showed relatively lower cell dry weight and produced less *mcl*-PHA, suggesting deletion of *tesA* did not promote *mcl*-PHA production (Fig. 4c, d). In contrast, KTc9n11 and KTc9n12 showed improvement in cell biomass (Fig. 4c, d). Moreover, KTc9n11 and KTc9n12, respectively, produced 29 and 140% more *mcl*-PHA than did KTc9n, up to 157 mg L$^{-1}$ (Fig. 4c and Supplementary Table 4). This revealed that deletion of either *tesB* or *tesBII* had positive effect on *mcl*-PHA accumulation, especially for *tesBII*.

In addition, it is known that *phaZ* (*mcl*-PHA depolymerase) knockout and *phaC* (*mcl*-PHA synthase) knockin enhance *mcl*-PHA accumulation in *P. putida*[19]. Therefore, *phaZ* was deleted and replaced by a copy of *phaC1*, generating the mutant KTc9n13 (Table 1 and Supplementary Table 3). Interestingly, a similar *mcl*-PHA titer was observed between KTc9n and KTc9n13, indicating that simultaneous *phaZ* elimination and *phaC1* insertion had little impact on PHA production (Fig. 4c). Therefore, an alternative strategy was examined, a single copy of a mutated *phaC1* gene[46] with its native promoter was inserted upstream of the *pha* gene cluster. The resulting mutant, KTc9n14, produced a higher *mcl*-PHA titer (95 mg L$^{-1}$) than KTc9n, confirming the role of this mutated *phaC1* in *mcl*-PHA production (Fig. 4c and Supplementary Table 4).

Finally, to further promote the *mcl*-PHA accumulation, the elements with positive effect, *tesB*, *tesBII* and mutated *phaC1*, were

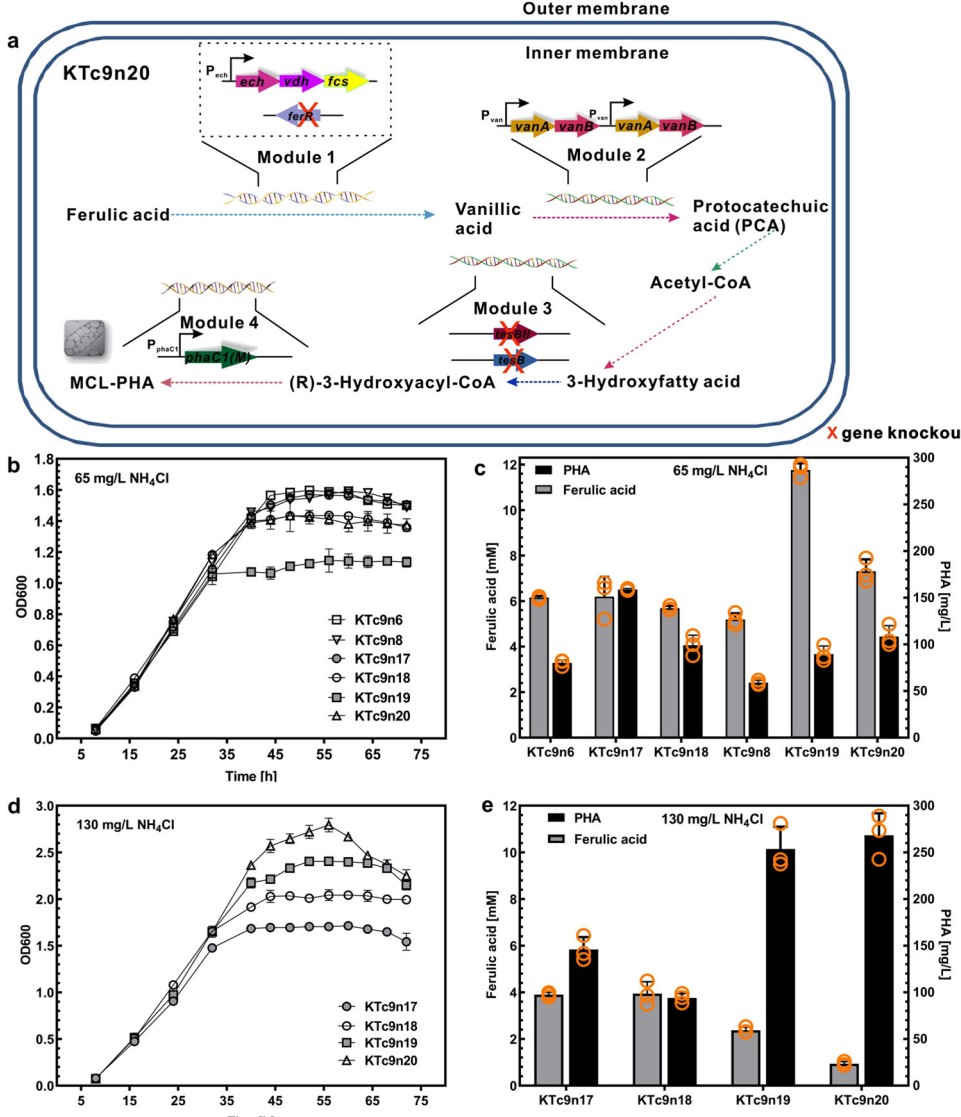

**Fig. 5 Metabolic engineering of ferulic acid to *mcl*-PHA bioconversion in *P. putida* KT2440. a** Metabolic engineering of ferulic acid-to-*mcl*-PHA bioconversion in *P. putida* KT2440. Four modules were integrated to synthesize *mcl*-PHA from ferulic acid in KT2440. **b**, **c** The cell growth, residual ferulic acid in medium, and *mcl*-PHA titer of the engineered strains under 20 mM ferulic acid-65 mg L$^{-1}$ NH$_4$Cl condition ($n = 3$ biologically independent experiments). **d**, **e** The cell growth, residual ferulic acid in medium and *mcl*-PHA titer of the engineered strains under 20 mM ferulic acid-130 mg L$^{-1}$ NH$_4$Cl condition ($n = 3$ biologically independent experiments).

integrated into the *P. putida* KTc9n strain. As a result, the double knockout mutant of thioesterases, KTc9n15 ($\Delta tesB\Delta tesBII$) and KTc9n16 ($\Delta tesB\Delta tesBII$ and *phaC1*), were constructed, respectively (Table 1). The accumulated *mcl*-PHA in KTc9n15 increased to 164 mg L$^{-1}$, while KTc9n16 did not further gains in *mcl*-PHA accumulation (105 mg L$^{-1}$, Fig. 4c and Supplementary Table 4).

**Integrated functional modules to improve ferulic acid-to-PHA conversion.** The aforementioned metabolic engineering addressed key metabolic steps for ferulic acid to *mcl*-PHA conversion. To integrate these functional modules, KTc9n6 and KTc9n8 were selected as the parent strains, where the *tesB* and *tesBII* were deleted with or without an integrated a copy of mutated *phaC1* (Fig. 5a). Consequently, four mutant strains were constructed, KTc9n17, KTc9n18, KTc9n19 and KTc9n20, respectively (Table 1). Cell growth, substrate utilization, and *mcl*-PHA titers of these four strains were assessed (Fig. 5b, c and Supplementary

Table 4). (i) In comparison with the parent strain KTc9n6, KTc9n17 and KTc9n18 exhibited similar cell growth. In contrast, KTc9n19 and KTc9n20 showed poorer cell growth than their parent strain KTc9n8. (ii) Compared to KTc9n6, KTc9n17 showed similar ferulic acid utilization, while KTc9n18 consumed more ferulic acid. Surprisingly, KTc9n19 and KTc9n20 utilized less ferulic acid than KTc9n8. (iii) KTc9n17 and KTc9n18 produced more *mcl*-PHA than KTc9n and KTc9n6, while KTc9n19 and KTc9n20 synthesized more *mcl*-PHA than KTc9n8. Among these four mutants, KTc9n17 accumulated the highest *mcl*-PHA titer, 158 mg L$^{-1}$ from 14 mM consumed ferulic acid, followed by KTc9n20, 108 mg L$^{-1}$ from 13 mM utilized ferulic acid.

**Optimized C/N ratio to further boost ferulic acid-to-PHA conversion.** Although the four integrated mutants produced more *mcl*-PHA than their parent strains, the substrate consumption was not further improved. Next, C/N ratio, which is

known to be essential in cell growth and *mcl*-PHA synthesis, was optimized. KTc9n and KTc9n8 showed the best performance in cell growth and substrate utilization with 20 mM ferulic acid and 260 mg $L^{-1}$ $NH_4Cl$, while KTc9n6 exhibited optimized cell growth and ferulic acid utilization in the presence of 130 mg $L^{-1}$ $NH_4Cl$ (Supplementary Fig. 3). The difference in strain performance between the two nitrogen concentrations (130 mg $L^{-1}$ and 260 mg $L^{-1}$) was small. Moreover, *Pseudomonas* species accumulate *mcl*-PHA under nitrogen starvation condition when non-fatty acid feedstock is utilized. Thus, 130 mg $L^{-1}$ $NH_4Cl$ was selected. First, cell growth of all four strains was improved under such condition, in contrast to that of under 65 mg $L^{-1}$ $NH_4Cl$. Higher cell biomass ($OD_{600}$ value and CDW) was obtained (Fig. 5b, d and Supplementary Fig. 3d). Among them, KTc9n20 reached the highest cell biomass, ~780 mg $L^{-1}$ (Supplementary Fig. 3d-4). Second, consistent with the improved cell growth, more ferulic acid was utilized by all four strains. KTc9n19 and KTc9n20 consumed more ferulic acid than KTc9n17 and KTc9n18, with KTc9n20 utilizing the most ferulic acid, ~ 20 mM. Third, correspondingly, KTc9n19 and KTc9n20 also produced a higher *mcl*-PHA titer, increasing to ~270 mg $L^{-1}$, whereas KTc9n17 and KTc9n18 produced *mcl*-PHA titers similar to that seen under 65 mg $L^{-1}$ $NH_4Cl$ condition (Fig. 5e and Supplementary Table 4). Taken together, KTc9n20, which integrated four functional modules, exhibited the best ferulic acid-to-PHA conversion via C/N ratio optimization, where almost 20 mM ferulic acid was utilized and 270 mg $L^{-1}$ *mcl*-PHA was accumulated (Fig. 5a, e and Table 4).

## Discussion

The gram-negative bacterium *P. putida* KT2440 plays an important role in biotechnological and biogeochemical applications[7,9,47]. Although several genome-editing approaches have been developed for *P. putida* KT2440[26,28,33], our CRP method proved to be an efficient, fast and convenient tool (Table 3). First, *pyrF* was employed as selection marker to allow efficient mutant screening, which is essential for targets with low mutation rate. Although recently reported CRISPR/Cas9-induced recombination methods show high efficiency (13–93.3% below 5 kb), it is highly variable among different genes, especially for large DNA fragment insertions or deletions[33]. To overcome this limitation, *pyrF* was mobilized to enable the genome-editing event selected by uracil prototrophy and following *pyrF* marker removal event selected by uracil auxotrophy and by resistance to 5-FOA. Variable genome-editing events were tested in this system, validating that all the mutation efficiencies were almost 100% (Supplementary Table 3). For long DNA fragment editing, which is always difficult in

genome editing, only an extension of the xylose induction time was required to ensure the efficiency. This suggested the mutation efficiency in our system theoretically reached 100%, which was independent of the genome location and DNA fragment length. In addition, it is noteworthy that the *pyrF* marker can be deleted, generating a scarless mutant, which is convenient for additional rounds of genome editing. Second, Cas9n was mobilized as an alternative strategy to alleviate lethality, reduce off-target effect, thereby improving editing efficiency. Off-target effect is often a challenge in the CRISPR/Cas9 system, as it not only decreases the mutation efficiency, but also leads to cell death due to multiple genome breaks[32]. λ-Red system, specific gRNA designed by CasOT and the low copy number vector (e.g., RK2 replicon, 5–7 copies per cell[48]) that carries Cas9 gene expression cassette was required to alleviate such damage[28]. In contrast, Cas9n, which generates single-nicks, is less toxic than Cas9[31]. Our study demonstrated that the *pyrF* deletion efficiency via CRISPR/Cas9n system is higher than that of CRISPR/Cas9 system (Table 2). Third, integrating CRISPR/Cas9n and λ-Red systems in the *P. putida* KT2440 genome decrease the size of the plasmid, increase the genetic stability, and hence make it easier to perform multiple genome manipulations. In fact, we initially tried to develop a one-plasmid system to simplify the process. However, unlike *E.coli* and *Clostridium cellulolyticum*[32,39], the three components, *cas9n*, sgRNA, and homologous repairing arms, in one plasmid are incompatible in *P. putida* (Supplementary Notes). Therefore, Cas9n and λ-Red recombination proteins were integrated into the bacterial chromosome to avoid plasmid instabilities. Fourth, the CRP strategy was also expanded to plasmid-curing application, as a fast, single-step plasmid-curing platform. Traditional methods are often tedious, requiring sequential rounds of growth under stressful or non-selective conditions to promote the appearance of plasmid-free segregants, consequently increasing the chance of accumulating unwanted mutations and are time consuming[49]. In our strategy, the functions of genome editing and plasmid curing are combined in one plasmid. It only requires different inducers to turn on the corresponding function and thus greatly shortening manipulation time and addressing this procedural bottleneck. Moreover, the *gfp* system was introduced, allowing us to easily identify non-fluorescent plasmid-cured colonies with an efficiency of 90% within 24 h (Supplementary Fig. 2b).

Although KT2440 has the ability to produce *mcl*-PHA from ferulic acid, the toxicity and low tolerance of ferulic acid, as well as the complex and variable PHA biosynthetic pathways, complicate the bioconversion process. Our study divided the nine genes known to be involved in the pathways into four modules, which facilitated the characterization of each aspect of the process, and consequently developed the engineered KTc9n20 strain

**Table 3 Comparison of different genetic editing tools in *Pseudomonas*.**

| Method | Time (days) | Scarless | Markerless | Insertion | Efficiency | Efficient plasmid curing | Number of plasmid | Organisms | References |
|---|---|---|---|---|---|---|---|---|---|
| CRP | 4 | Yes | Yes | Yes | 100% | Cas9n-mediated plasmid curing | 2 | *P. putida* KT2440 | This study |
| CRISPR/cas9-λ Red | 5 | Yes | Yes | Yes | 70–100% | Cas9-mediated plasmid curing | 2 | *P. putida* KT2440 | 28 |
| CRISPR/cas9-Ssr | 4–5 | Yes | Yes | limited | 13–93% | without antibiotic selection | 3 | *P. putida* KT2440 | 33 |
| I-SceI | 5–6 | Yes | Yes | ND | 14–81% | suicide vector/ SacB | 2 | *P. putida* KT2440 | 23 |
| Flp-FRT | 5–6 | FRT site (48 bp) | Yes | ND | ND | without antibiotic selection | 2 | *P. aeruginosa* POA1 | 10 |
| λ-Red-Cre/loxP | 4–6 | LoxP site (34 bp) | Yes | ND | 70–100% | suicide vector/ SacB | 1 plus Linear DNA | *P. putida* KT2440 | 7 |
| Homologous recombination with *pyrF* marker | 14 | $Gm^R$ site | No | ND | ND | suicide vector | 1 | *P. putida* KT2440 | 24 |

*ND* not determined.

**Table 4 *mcl*-PHA production from aromatic compounds in *Pseudomonas* strains.**

| Carbon Source (Ini-Con mM)[a] | Nitrogen Source (mg L$^{-1}$) | Cultivation mode | CDW (mg L$^{-1}$) | PHA titer (mg L$^{-1}$) | Measurement method for PHA production | Organisms | References |
|---|---|---|---|---|---|---|---|
| Ferulic acid (20–15.74) | NH$_4$Cl (130) | Shake flask batch | 540 | 59.3 | GC | *P. putida* KT2440 | This study |
| Ferulic acid (20–19.06) | | | 780 | 268.3 | | *P. putida* KTc9n20 | |
| Ferulic acid (20–15.74) | NH$_4$Cl (65) | Shake flask batch | 540 | 65.4 | GC | *P. putida* KT2440 | |
| | | | | 105.0 | Nile Red staining | | |
| | | | | 218.0 | Weighing | | |
| Ferulic acid (10–10) | ND[b] | Shake flask batch | 436 | 170.0 | Flow cytometry via Nile Red staining | *P. putida* KT2440 | 9 |
| *p*-coumaric acid (12–12) | | | 470 | 160.0 | | | |
| Vanillic acid (15–ND) | NH$_4$Cl (65) | Shake flask batch | 715 | 246 | Weighing | *P. putida* Axyl_alkKphaGC1 | 13 |
| *p*-coumaric acid (12–12) | (NH$_4$)$_2$SO$_4$ (130) | Fed-batch, flask | 483 | 241.0 | GC/MS | *P. putida* AG2162 | 19 |
| *p*-coumaric acid (79.3-<4.5) | (NH4)2SO4(528) | Fed-batch, flask, HCD[c] | 1758 | 953 | | | |
| Phenylacetic acid (15–15) | NaNH$_4$PO$_4$ 4H$_2$O (67) | Shake flask batch | 824 | 247.0 | Waters gel permeation chromatograph | *P. putida* CA-3 | 53 |
| Styrene (19–ND[b]) | | | 796 | 195.0 | | | |
| Styrene (20–ND) | NH$_4$Cl (25) | Solid-state fermentation | 530 | 140.0 | GC | *P. putida* mt-2 | 54 |
| Benzene (20–ND) | | | 340 | 48.0 | | *P. putida* F1 | |
| Benzene (11–ND) | NH4Cl (500) | Fed-batch fermentation | 290 | 2.9 | GC/MS | *P. fulva* TY16 | 55 |
| Toluene (20–ND) | NH4Cl (25) | Solid-state fermentation | 370 | 81.0 | GC | *P. putida* mt-2 | 54 |
| | | | 720 | 160.0 | | *P. putida* F1 | |
| Toluene (9–ND) | NH4Cl (500) | Fed-batch fermentation | 290 | 16.2 | GC/MS | *P. fulva* TY16 | 55 |
| Ethylbenzene (20–ND) | NH4Cl (25) | Solid-state fermentation | 670 | 98.0 | GC | *P. putida* F1 | 54 |
| Ethylbenzene (8–ND) | NH4Cl (500) | Fed-batch fermentation | 260 | 8.8 | GC/MS | *P. fulva* TY16 | 55 |

[a]Ini-Con mM, initiate carbon substrate concentration - consumed carbon substrate concentration (mM).
[b]ND, not determined.
[c]HCD, high-cell-density.

with greatly improved ferulic acid-to-*mcl*-PHA conversion (Table 4). On one hand, Gene expression ratio is key in ferulic acid catabolism optimization. Enhancing gene expression levels, via transcriptional regulation or increasing gene copies, is a known to improve substrate utilization[50], which was validated in KTc9n3 and KTc9n6. However, it is likely to cause the accumulation of toxic intermediate metabolites and redox imbalance. Ferulic acid degradation by KTc9n7 did not outperform that of KTc9n6 is an example (Fig. 3d). To counter this hurdle, gene expression ratio was optimized in the advanced strain KTc9n8 to exhibit the highest ferulic acid consumption and lowest vanillic acid accumulation. On the other hand, numerous studies have attempted to design metabolic engineering strategies to increase metabolic flux from fatty acid synthesis to PHA biosynthesis[51−55]. However, the effects on *mcl*-PHA accumulation are variable under different conditions, such as different feedstock. Thereby, these strategies must be examined in a trial-and-error type of approach (Table 4). (i) Deletion of thioesterases were reported to boost *mcl*-PHA production in *E. coli* when grown on glucose[43], as well as in *Alcanivorax borkumensis* SK2 with alkanes as carbon source[45]. TesA in *E.coli* was reported to preferably cleave acyl-CoA of C12-C18, while TesB is known to cleave C6-C18 acyl-CoA esters[43,45]. Deletion of *tesB* in KTc9n11 had positive effect, while deletion of *tesA* in KTc9n10 had a negative effect on *mcl*-PHA accumulation. Considering that accumulated *mcl*-PHAs have a heterogeneous monomer composition, with 6–12 carbon atoms (Supplementary Table 4), such different effects are probably due to the different substrate specificities of the thioesterases. In addition, it is interesting to note that deletion of *tesBII* substantially stimulated *mcl*-PHA accumulation, reaching 157 mg L$^{-1}$, whereas PHA titer in KTc9n11 ($\Delta tesB$) was 84 mg L$^{-1}$. This revealed the distinct functions of the TesB and TesBII in *P. putida* KT2440, with the latter apparently specifically acting on 3-hydroxyacyl-CoA esters[45] (Fig. 4a). (ii) PHA metabolism is a bidirectional dynamic process in which there is a continuous cycle of synthesis and degradation[56]. It was speculated that removing the PHA degradation gene and strengthening the PHA biosynthesis gene could improve PHA production. However, the impact on PHA yield by elimination of PHA depolymerase ($\Delta phaZ$) in *Pseudomonas* mutants are debatable. Some studies reported a

positive effect in *P. putida* strain U[57], *P. putida* KT2442[58] and *P. putida* AG2228[19], while others suggested little impact in *P. resinovorans* and *P. putida* KT2440[56]. This discrepancy maybe due to genetic (e.g., differences in genotype of investigated strains and the deleted *phaZ* DNA sequences) and environmental (e.g., culture conditions) variables. In our study, the KTc9n13, in which *phaZ* was deleted and replaced by *phaC1*, did not exhibit significant difference in *mcl*-PHA accumulation ($p = 0.48$) and biomass ($p = 0.13$) with KTc9n (Fig. 4c). As *phaZ* and *phaC* are important in maintaining PHA metabolism to balance carbon resources[59], simultaneous *phaZ* disruption and *phaC1* insertion might break the balance, although an insertion of the mutated *phaC1* resulted in higher PHA titer (Fig. 4c, d). Thus, further investigation and optimization of the PHA regulation mechanism, including regulators and the *cis*-elements of genes involved in PHA synthesis cluster, is required.

In conclusion, this study developed a CRP genome-editing strategy. It maintains cell viability, increases mutation efficiency, simplifies manipulation, and avoids unwanted mutation accumulation. Through this approach, we genetically investigated ferulic acid-to-PHA bioconversion in detail and further constructed the engineered *P. putida* KTc9n20 strain to successfully expand ferulic acid valorization, in addition to its well-known ferulic acid-to-vanillin biotransformation. Together with a microbial lignin depolymerization system under development[5], exploration of PHA regulation mechanism under aromatic compounds, and further improvements in culture conditions in the near future, could provide insights into microbial carbon sink as well as improving lignin-consolidated bioprocessing schemes.

## Methods

**Bacterial strains and culture conditions.** The bacterial strains used in this study are shown in Table 1. *Escherichia coli* DH5α strain (TransGen Biotech) was used for all molecular manipulations in plasmid construction and was grown on Luria-Bertani (LB) medium at 37 °C. *P. putida* KT2440 strain (ATCC 47054) was purchased from the American Type Culture Collection (ATCC). For genome editing, KT2440 strain and its mutants were grown on either LB medium or 15 mM glucose-M9 minimal medium at 30 °C[60]. For *mcl*-PHA production, *P. putida* strains were cultivated in 20 mM ferulic acid-M9 mineral medium supplemented with 65–1000 mg L$^{-1}$ NH$_4$Cl and 20 μg mL$^{-1}$ uracil with a shake speed of 200 rpm at 30 °C[13]. Cell growth was spectrophotometrically monitored by measuring the optical densities at 600 nm (OD$_{600}$).

**Plasmid construction**. All plasmids used in this study are summarized in Table 1, while the primers and N20 sequences followed by the PAM used in this study are listed in Supplementary Table 1. The *cas9n* gene from *Streptococcus pyogenes* SF370[61] was codon optimized and synthesized with a His tag-encoding sequence at the C terminus by Suzhou Synbio Technologies Company (Suzhou, China). All N20NGG sites were extracted as previously described[39]. Plasmids were constructed according to the standard molecular cloning protocols[60]. All further details regarding the sgRNA design and vector construction are provided in Supplementary Methods. All plasmids were verified by DNA sequencing for further studies.

**CRP Genome-editing procedures**. To integrate the $P_{min}::cas9n$ and $P_{xylA}::gam$-*bet-exo* expression cassettes and replace the *pyrF* gene, **pBBR1–2** was transformed into KT2440 through electroporation. The strain was cultured at 30 °C in LB medium supplemented with 50 µg mL$^{-1}$ kanamycin and 20 µg mL$^{-1}$ uracil for 30 h. Meanwhile, 4 mM xylose was added to LB medium at the initial stage to induce the expression of λ-Red recombinase. Next, appropriately diluted cells were plated on LB selective agar plate (Ura$^+$+5-FOA + Km) to select the mutant strain KTc9n (Fig. 1, Table 2 and Supplementary Table 3). Twenty colonies were randomly picked for further PCR screening and DNA sequencing to identify KTc9n.

To delete *icd*, the **pBBR1-3** was transformed into KTc9n by electroporation. After 18 h cultivation at 30 °C in liquid LB selective medium (xylose+Ura$^+$+Km) for genome editing, 100 µL cells were transferred to 5 mL fresh 15 mM glucose-M9 minimal medium (Ura$^-$), supplemented with 4 mM vanillic acid to induce the expression of gRNA targeting the palindromic sequence of **pBBR1-3** plasmid, and grown overnight at 30 °C with shaking at 200 rpm for plasmid curing. Appropriately diluted cells were subsequently plated on M9 minimal agar plates (Ura$^-$), where the colonies without green fluorescence indicated that **pBBR1-3** had been eliminated (Fig. 1). Consequently, twenty non-fluorescent colonies were randomly picked for PCR screening and DNA sequencing, generating the mutant KTc9nΔ*icd*, which contained the selection marker *pyrF* (Table 2 and Fig. 1). The *pyrF* marker was removed by the next round of genome editing, where **pBBR1-4** was transformed to KTc9nΔ*icd*, generating the mutant KTc9n9.

For *tesB* and *tesBII* double-gene disruption, **pBBR1-7** was transformed into KTc9n to delete *tesB*, while **pBBR1-8** was subsequently introduced to remove *pyrF* marker, generating the mutant KTc9n11. In the following round of genome editing, **pBBR1-19** and **pBBR1-20** were sequentially transformed into KTc9n11 to knockout *tesBII*. Consequently, KTc9n15 was constructed. For *phaZ* deletion and *phaC1* insertion, **pBBR1-10** was transformed into KTc9n to delete *phaZ* and *phaC1*. Then, pBBR1-11 was introduced to replace *pyrF* maker with two copies of *phaC1*, generating the mutant KTc9n13 strain. Additional genomic mutants in this study were constructed according to the same procedures (Table 1, Supplementary Table 3 and Supplementary Methods).

**Determination of editing efficiency**. The *pyrF* gene deletion efficiency was calculated as previously described[62]. Briefly, each of the plasmids (1 µg) was transformed into KT2440 by electroporation. Positive mutant colonies were selected by the LB selective (Ura$^+$+5-FOA + Km+Gm) plate, while the total viable cells were grown on LB ((Ura$^+$+Km+Gm) plate. Positive colonies were further verified by PCR and DNA sequencing. The editing efficiencies of *pyrF* gene via different genome-editing strategies were calculated as the number of positive colonies as a percentage of the total number of colonies on plate (Supplementary Methods).

In addition, for the editing efficiency via the CRP method, 8–20 colonies were randomly picked from the LB selective plate (xylose+Ura$^+$+Km). Positive colonies were verified by PCR and DNA sequencing. The gene editing efficiency was calculated as the number of positive colonies as a percentage of the total picked colonies. All of the experiments were performed in technical triplicate.

**RNA isolation and RT-PCR**. RNA isolation and RT-PCR was performed as previously described[13]. Briefly, 1 mL *P. putida* KT2440 cell culture in LB medium was harvested during the mid-exponential phase (OD$_{600}$, ~5.0). The total RNA was isolated using a TransZol Up Plus RNA kit (Transgen Biotech) and then reverse transcribed using TransScript one-step genomic DNA (gDNA) removal and cDNA synthesis supermix kit (Transgen Biotech). The cDNA product was diluted as appropriate and utilized as a template. *pyrF* gRNA expression was examined by reverse transcription-PCR (RT-PCR) using 16 S rRNA gene as an internal calibrator.

**SDS-PAGE analysis**. To examine the expression of full-length His-tagged Cas9n in *P. putida* KT2440, a single colony of each, pCas9n or pPROBE-GT (empty vector), transformants was cultivated in LB medium. Cells were lysed in the SDS loading buffer, and then supernatant cell lysates were subjected to 12% gel SDS-PAGE separation. In addition, Cas9n protein was purified by Ni-NTA column (Takara) and separated by SDS-PAGE as previously described[5].

**Batch cultivation in shake flasks**. A single colony of each *P. putida* KT2440 mutant was inoculated into 5 mL LB selective medium (Ura$^+$+5-FOA) and cultivated at 30 °C and 200 rpm for 24 h. Subsequently, 0.2 mL cell culture was transferred into 20 mL 20 mM ferulic acid -M9 mineral medium (Ura$^+$) at 30 °C, 200 rpm.

When the stationary phase was reached, 1% (vol/vol) of seed inoculum was incubated in 100 mL of 20–40 mM ferulic acid-M9 mineral medium (Ura$^+$) at 30 °C, 200 rpm. After 48 h cultivation (at the stationary phase), 100 mL cell culture was harvested by centrifugation (10 min, 8000 rpm, 4 °C), washed twice with 15 mL Nanopure water, lyophilized (Lyophilizer Alpha 1–4 LSC; Martin Christ Gefriertrocknungsanlagen GmbH, Osterode am Harz, Germany) at −59 °C and 0.140 × 10$^5$ mPa for a minimum of 24 h, and weighed. The lyophilized cells were kept at −80 °C for PHA extraction. For analysis of residual ferulic acid and vanillic acid in the medium, 2 mL of supernatant from each culture was collected and stored at −80 °C until HPLC analysis. All of the experiments were performed in biological triplicate.

**Analytical methods**. PHA was extracted and purified from each lyophilized cell sample using the hot chloroform method as previously described[17]. For PHA production and composition analysis, each purified PHA was acidic methanolysis and assayed using a gas cheomatograph (Agilent 7890 A GC with FID detector; Santa Clara, CA) according to previously described protocols[58]. PHA production of KTc9n and KT2440 were also determined by Nile red staining and weighing as previously described[9,17,63].

The concentrations of ferulic acid and vanillic acid in the medium were measured by HPLC, as previously described[8]. Briefly, each culture supernatant was diluted 1:20 with 0.2 % acetic acid and analyzed by HPLC using a Thermo UltiMate 3000 HPLC system equipped with Thermo Acclaim 120 C18 (4.6 mm × 250 mm, 5 µm) column. The flow rate was 1 mL per min and the absorbance was measured at 231 nm for 25 min. All of the samples were prepared and analyzed in biological triplicate.

The NADH/NAD$^+$ ratio was measured through NADH/NAD$^+$ Assay Kit (SCBK-END100, Chengdu Xinchuang Boke Biotechnology Co., Ltd.) as previously described[64]. Briefly, 1 mL of the cell culture was harvested by centrifugation (10 min, 8000 rpm, 4 °C) at the stationary phase. Cell pellets were washed with PBS buffer and resuspended with 100 µL NADH/NAD$^+$ extraction buffer. And then NADH/NAD$^+$ was detected according to the kit manual.

**Plasmid copy number calculation**. Quantitative real-time PCR (qPCR) was performed to measure the copy number of the plasmids in *P. putida* KT2440 (Table 2). Kanamycin resistance gene (*nptII*) in pBBR1-1 and gentamicin resistance gene (*aacC1*) in pGT-series plasmids, as the single copy genes of the plasmids, were utilized as the target gene to represent the copy number of plasmids. In addition, the housekeeping RNA polymerase sigma σ$^{70}$ factor (*rpoD*) in KT2440 chromosomal was employed as the reference gene. First, KT2440 strains harboring corresponding plasmids were collected when they were grown to stationary phase in LB medium at 30 °C. Then the genomic DNA and plasmid DNA were extracted, respectively. Second, the DNA fragments of *nptII*, *aacC1* and *rpoD* were amplified, purified and serially diluted as the templates ($1 \times 10^5 \sim 1 \times 10^9$ copies µL$^{-1}$) to construct the standard curves. Third, qPCRs were performed using the SYBR Premix Ex Taq II Kit (TakaRa) with the real-time PCR detection system (Roche, LightCycler 480) to construct the standard curves and measure the absolute copy numbers of *nptII*, *aacC1* and *rpoD*. The primers used in qRT-PCR are listed in Supplementary Table 1. The plasmid copy number was calculated using the following equation: plasmid copy number = copy number of target gene in plasmid/ copy number of *rpoD* in genome. All of the samples were prepared and analyzed in biological triplicate.

**Statistics and reproducibility**. All growth experiments, analytical experiments (e.g., *mcl*-PHA titers, the concentrations of ferulic acid and vanillic acid, and the NADH/NAD$^+$ ratios), and plasmid copy number assessment were performed in biological triplicate. Mean from three independent biological experiments was presented in each plot, in which error bars represented standard deviation. A number *n* suggested biological replications.

Unpaired two-tailed Student's *t* test was used for comparison of two experimental groups. For all analyses, $p < 0.05$ are considered significant.

**Reporting summary**. Further information on research design is available in the Nature Research Reporting Summary linked to this article.

## Data availability

All data generated or analyzed during this study are included in this published article. Source data underlying the graphs and charts are provided in Supplementary Data 1–5.

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

## Acknowledgements

This work was supported by the National Natural Science Foundation of China (9195116 and 41606154) and National Key Research and Development Project (2019YFA0606704). The funders had no role in the study design, data collection and analysis, decision to publish, or preparation of the manuscript.

## Author contributions

L.L. and Y.Z. conceived and designed the experiments. Y.Z. performed the experiments. L.L. and Y.Z. analyzed the data. L.L., J.Z., and N.J. wrote the paper. H.W. and Z.Z. contributed reagents and analysis tools. All authors read and approved the final manuscript.

## Competing interests

The authors declare no competing interests.
