## [Peer Review File · Communications Biology]

Reviewers' comments:

Reviewer #1 (Remarks to the Author):

Review:

In this study, the authors have developed CRISPR-related genome editing tools, combining λ -Red recombination system and pyrF selection (positive and negative) marker, resulted with a fine-tuned genomic editing tool in *P. putida* KT2440 and its derivative strains, allowing high-efficiency gene deletion and insertion, with an efficiency of 100% for all targets. By using the high-efficiency method, the FA metabolism was optimized to allow cell growth under high FA concentrations, to produce acetyl-CoA, followed by an engineered PHA synthesis pathway, resulting in high PHA content relative to the wild-type strain. Generally, this study tells a complete story of metabolic engineering in *P. putida* to produce PHA from aromatic compounds, including developing new tools, module regulation, and optimization of cultivation strategy. However, there remain many details need to be stressed and declared.

Concerns:

1. The author chose Cas9 nickase (nCas9) instead of Cas9 nuclease, claiming that Cas9 nuclease cause lethality to cells. While in the cited publication (Page 7 line 155; Sun J. et al., 2018), Cas9 nuclease seems to be less toxic to the cells. One possible reason is that Sun J. et al. use a low-copy-number RK2 replicon, while the replicon in this study is the high-copy-number P15A replicon (the plasmid details were not presented here and I found it on Addgene). The authors should address this.
2. A follow-up concern, the details of plasmids are not presented here, especially the replicon. Considering the different copy numbers in various bacteria, the plasmid copy numbers should also be listed when making comparison of editing efficiency.
3. The authors inserted nCas9 and λ -Red recombinase into the genome to achieve a single-plasmid system, while raising a problem that overexpression of those heterologous genes may influence the cell growth. The author should address this issue, and include data to support it.
4. In the part of FA metabolism regulation, the FA utilization rates were increased and VA accumulation rates were reduced by overexpressing vanAB (Page 12, line 290-291). The results are impressive, while it may not be comprehensive enough to claim that the improved behaviors benefited from cofactor balance. At least the authors should present clear data to support the statement.
5. The authors need to be more aware of previous work in this area. The concept of lignin consolidated bioprocessing was proposed in a Green Chemistry paper (Salvachua et al.) from 2015 and following work from Professor Yuan at Texas A&M University. Production of PHAs

from engineered strains of *P. putida* using hydroxycinnamic acids was recently also published by the Guss group in 2019 in *Microbial Biotechnology*.

6. Generally, how is this work “consolidated bioprocessing”? Consolidated bioprocessing implies using a microbe to secrete enzymes that cleave polymers, followed by microbial uptake of monomers, and production of a target product. Conversion of ferulic acid (an aromatic “monomer”) is more appropriately termed “aromatic bioconversion” or something similar.

7. The authors also discuss vanillate as the “central node” for FA conversion. There are several pathways to catabolize FA that have been described. I would just remove this statement, as vanillate is certainly not the “central node” for PHA production from aromatic compounds such as ferulic acid (what does “central node” even mean?). The entire paragraph motivating FA-to-VA is quite odd and should be reworked.

8. The use of the word “novel” in the scientific literature is hackneyed at this point. By definition, original research reports should be novel, so needing to claim this is redundant. This should be removed throughout the paper.

9. Previous work has also been done on overexpression of VanAB to overcome bottlenecks in VA catabolism, including work from Yuan et al. and Salvachua et al. in *P. putida* KT2440. This work is not discussed or mentioned.

Minor concerns:

1. Page 4, line 79. The drawbacks of specific strains are due to different engineering approaches and purposes, it’s improper to compare specific construction to one whole species here, it’s not an apple to apple comparison.

2. Page 4, line 85. The statement of here is not accurate and misleading, the disadvantages here should better be addressed respectively.

3. Page 4, line 89-91. The statement here is not accurate, there are many Cas9-based genome editing methods, various base editors for example. Better replace Cas9-based with Cas9-induced recombination methods.

4. Page 5, line 101. Replace ‘limiting their’ with ‘limiting its’.

5. λ -Red or λ -red system? Please go thoroughly and correct.

6. Page 21, line 507. Change to “trial-and-error”.

7. Page 21, line 511. Change “under gluconate” to “with gluconate as carbon source”.

8. Page 24, line 568. Should be ‘rpm’, no blanks.

9. Page 27, line 660. 'x' change to 'x'.
10. Page 28, line 674 for example. There should be a blank between number and unit, please go thoroughly and correct.
11. Page 31, line 745, please correct the error.
12. Please correct all the errors in the References part, capitalize each word or the first word?
13. Page 34, line 825, phaZ should be italic.
14. Supplementary Page 1, line 5, please correct the error.

Reviewer #2 (Remarks to the Author):

This manuscript describes the development of a highly efficient CRISPR/Cas9n genome editing strategy (CRP) in *Pseudomonas putida*. Also reported is the application of CRP for improving FA-to-PHA bioconversions. The experiment data is substantial and analyzed deeply.

Major

1. L77-78 mentions 'avoiding the drawbacks (e.g., inefficient heterologous gene expression and plasmid instabilities) of recombinant strains, such as *Escherichia coli*'. The description of drawbacks for the recombinant *Escherichia coli* is not persuasive.
2. L115 mentions the expression of nine genes that are divided into four engineered modules. However, Fig 5(A) showed eight genes were integrated into genome and two genes were deleted.
3. L124 In this study, more than 20 strains were constructed to improve mcl-PHA production. Which strains would produce 270mg/L mcl-PHA? This section needs more description.
4. L131 This study first tested the reported two-plasmid CRISPR/Cas9 system and a CRISPR/Cas9- λ -red system in *P. putida* KT2440, respectively. This section is vague and difficult to understand. If you had tested this reported system, please add more description.
5. L228 Four different length genes or gene clusters were integrated into the chromosome of KTC9n. Which gene loci were used for the integration of these different length genes or gene cluster? How many colonies were picked from the plates for identification in each gene integration experiment? These details are very important for a highly efficient genome editing system.
6. L285 In addition, compared to KTC9n, the cell growth of KTC9n4 was also improved, which should be the result of more utilized VA, as a carbon source, to support the growth KTC9n4, in addition to FA. Why would vanillic acid(VA) be used as a carbon source? Please add the relating reference.
7. L433 variable genome editing events, including single-gene deletion, double-gene

disruption, and large DNA fragment integration (>10 kb), respectively, were tested in this system. I can't find the double-gene disruption data in this paper. Please provide more accurate information.

8. L624 'each of the plasmids (1 µg) were transformed into KT2440 by electroporation' should be 'each of the plasmids (1 µg) was transformed into KT2440 by electroporation'.

Reviewer #3 (Remarks to the Author):

Authors develop a genome editing strategy (CRP) for *P. putida* KT2440 using an integrated CRISPR/Cas9n-λ red system with pyrF as a selection marker, which maintains cell viability and strain genetic stability, increases mutation efficiency, and simplifies manipulation . Four functional modules consisting of nine genes involved in FA catabolism and polyhydroxyalkanoate (PHA) biosynthesis, were integrated into the KT2440 genome to improve FA-to-PHA bioconversion. The study establishes a simple and highly efficient genome editing strategy. A few issues need be addressed:

1. *P. putida* KT2440 can already produce many grams PHA. This study showed only mg level PHA. Something must be wrong with the growth and PHA production;
2. What kind of PHA was produced, scl- or mcl PHA? What mcl PHA was produced? Not show at all.

Before these issues are properly addressed, this paper can not be accepted.

Reviewers' comments:

Reviewer #1 (Remarks to the Author):

*In this study, the authors have developed CRISPR-related genome editing tools, combining λ -Red recombination system and pyrF selection (positive and negative) marker, resulted with a fine-tuned genomic editing tool in *P. putida* KT2440 and its derivative strains, allowing high-efficiency gene deletion and insertion, with an efficiency of 100% for all targets. By using the high-efficiency method, the FA metabolism was optimized to allow cell growth under high FA concentrations, to produce acetyl-CoA, followed by an engineered PHA synthesis pathway, resulting in high PHA content relative to the wild-type strain. Generally, this study tells a complete story of metabolic engineering in *P. putida* to produce PHA from aromatic compounds, including developing new tools, module regulation, and optimization of cultivation strategy. However, there remain many details need to be stressed and declared.*

Concerns:

1. *The author chose Cas9 nickase (nCas9) instead of Cas9 nuclease, claiming that Cas9 nuclease cause lethality to cells. While in the cited publication (Page 7 line 155; Sun J. et al., 2018), Cas9 nuclease seems to be less toxic to the cells. One possible reason is that Sun J. et al. use a low-copy-number RK2 replicon, while the replicon in this study is the high-copy-number P15A replicon (the plasmid details were not presented here and I found it on Addgene). The authors should address this.*

Response: Yes. We have performed the experiments to calculate the copy number of vectors that were used for genome editing. The copy number of pVS1 replicon vectors that carry the CRISPR/Cas9n expression cassette was 8~15 copies/cell (Table 3).

We have modified the text as suggested across the manuscript. The text now reads (*see Page 7-8, Line 168-175*):

“Moreover, considering that various vectors with different copy numbers were used to express the four components (Cas9, the gRNA cassette, λ -Red system and the homologous repairing arms) between this study and previous work ¹, expression dose of the four functional elements was also a factor contributing to the varied mutation rates (Table 3 and Table S2). Therefore, highly variable editing efficiency of reported CRISPR/Cas9 systems restricts whole genome editing in bacteria in general, and certain genes of *P. putida* in particular, which have either low repairing ability or poor mutation rate.”

Another example of our revision is (*see Page 20, Line 479-483*):

“Off-target effect is often a challenge in the CRISPR/Cas9 system, as it not only decreases the mutation efficiency, but also leads to cell death due to multiple genome breaks ². λ -Red system, specific gRNA designed by CasOT and the low copy number vector (e.g., RK2 replicon, 5-7 copies/cell ³) that carries Cas9 gene expression cassette was required to alleviate such damage ¹.”

2. *A follow-up concern, the details of plasmids are not presented here, especially the replicon. Considering the different copy numbers in various bacteria, the plasmid copy numbers should also be listed when making comparison of editing efficiency.*

Response: Yes. As our response to Review-Comment No. 1 above, we have performed the experiments to calculate the plasmid copy number and provided the detailed information in Table 2 (for replicon) and Table 3 (for copy number). Please refer to these tables in the revised version.

3. *The authors inserted nCas9 and λ -Red recombinase into the genome to achieve a single-plasmid system, while raising a problem that overexpression of those heterologous genes may influence the cell growth. The author should address this issue, and include data to support it.*

Response: Yes. KTc9n does not show obvious growth difference with KT2440 (Figure 2B and Figure S2A). We have revised the text accordingly.

The text now reads (*see Page 9, Line 210-212, Figure 2 and Figure S2A*): “Therefore, KTc9n, which showed no obvious growth difference from KT2440, was used as the host strain for the following genome editing (Figure 2 and Figure S2A).”

Another example of our revision is (*see Page 11, Line 263-268*): “Thus, *P. putida* KT2440 and KTc9n were cultured in 20 mM ferulic acid-M9 mineral medium with 65 mg/L NH₄Cl, respectively. Both strains showed similar cell growth, in which ~10 mM ferulic acid was consumed and ~65 mg/L *mcl*-PHA titer was accumulated (Figure 2B). No significant difference in ferulic acid utilization, NADH/NAD⁺ ratio and *mcl*-PHA production between KTc9n and KT2440 were observed ($p > 0.05$, Figure 2B and Table S4).”

4. *In the part of FA metabolism regulation, the FA utilization rates were increased and VA accumulation rates were reduced by overexpressing vanAB (Page 12, line 290-291). The results are impressive, while it may not be comprehensive enough to claim that the improved behaviors benefited from cofactor balance. At least the authors should present clear data to support the statement.*

Response: Yes. We have performed experiments which have fully addressed this excellent question. The corresponding section has been completely re-written to reflect these new experimental results.

The text now reads (*see Page 12-14, Line 294-338; Figure 3D*):

“To further enhance the FA utilization, these two strategies were integrated. The *ech-vdh-fcs* gene cluster was inserted into the genome of KTc9n3 and the resulting mutant was designated KTc9n4 (Table 2). However, neither KTc9n4 cell growth or FA consumption were further improved, in comparison to KTc9n1 (Figure 3C-D). Fcs, Ech and Vdh enzymes convert FA to vanillic acid (VA), whereas NAD⁺ is consumed and NADH is generated during this reaction (Figure 3B)⁴. To reveal whether this limitation was caused by intermediate metabolite accumulation and/or redox imbalance, the concentrations of VA and the ratio of NADH to NAD⁺ of these mutants were assessed. The concentration of VA, as generated by KTc9n1, KTc9n3 and KTc9n4, respectively, was higher than that of KTc9n. Of particular note, KTc9n4 attained the highest level (~4.8 mM, Figure 3D). Similarly, the NADH/NAD⁺ ratios of them were higher than that of KTc9n, especially for KTc9n1 and KTc9n4. Together, this indicated KTc9n4 accumulated more VA and consumed more NAD⁺ cofactor, resulting in insufficient redox for energy metabolism, e.g., TCA cycle (Figure 3B)⁴. A previous study suggested that overexpression of *vanAB* promotes VA catabolism in *P. putida* A514⁵, although

their role was not reported in *P. putida* KT2440. Therefore, we inferred that increasing the expression of *vanAB*, which catabolizes VA and regenerates NAD⁺ from NADH, should counter this hurdle (Figure 3B).

To explore this idea, KTc9n5, carrying a second copy of *vanAB* genes, was constructed (Table 2). Compared to KTc9n, it showed half the vanillic acid concentration, lower NADH/NAD⁺ ratio and more consumed FA (Figure 3D). Moreover, the cell growth of KTc9n5 was also improved (Figure 3C). These results suggested *vanAB* plays a role in alleviating VA accumulation and NAD⁺ imbalance. Thus, one copy of *vanAB* genes was inserted into the chromosomes of KTc9n1 and KTc9n4, generating the strains KTc9n6 and KTc9n7, respectively (Table 2). As expected, KTc9n6 exhibited greater FA utilization (~14 mM) and accumulated less VA (~1 mM) with lower NADH/NAD⁺ ratio, as compared to KTc9n1 (Figure 3D). Moreover, its cell growth was greatly improved, validating our hypothesis (Figure 3C). On the other hand, the performance of KTc9n7 on FA utilization and cell growth was improved, and NADH/NAD⁺ ratio decreased, in contrast to KTc9n4 (Figure 3D). However, KTc9n7 didn't outperform the KTc9n6 strain, exhibiting relatively poorer cell growth and FA utilization (Figure 3C and D). It was speculated that this might have been the result of an inappropriate gene expression ratio. Unlike KTc9n6, simultaneous single copy *ech-vdh-fcs* gene cluster insertion and *ferR* gene deletion in the genome of KTc9n7 should greatly accelerate the FA to VA conversion, whereas a single copy *vanAB* genes insertion might be insufficient for VA catabolism and cofactor requirement. This speculation was validated when, in contrast to KTc9n6, higher concentration of VA and NADH/NAD⁺ ratio were observed in KTc9n7 (Figure 3D). Therefore, optimizing the ratio of *ech-vdh-fcs*, *vanAB* and *ferR* gene copies appears to have the potential to address this bottleneck. Consequently, two copies of the *vanAB* genes were inserted into the KTc9n4 genome, constructing the mutant KTc9n8 (Table 2). As expected, KTc9n8 showed the best performance in cell growth and FA consumption (~15 mM) among the mutants, which was consistent with its lowest concentration of vanillic acid (~0.8 mM) and lower NADH/NAD⁺ ratio (Figure 3C-D)."

5. *The authors need to be more aware of previous work in this area. The concept of lignin consolidated bioprocessing was proposed in a Green Chemistry paper (Salvachua et al.) from 2015 and following work from Professor Yuan at Texas A&M University. Production of PHAs from engineered strains of P. putida using hydroxycinnamic acids was recently also published by the Guss group in 2019 in Microbial Biotechnology.*

Response: Yes. We have cited these literatures. Please refer to **Page 3-4, Line 70-82** in the revised version.

6. *Generally, how is this work "consolidated bioprocessing"? Consolidated bioprocessing implies using a microbe to secrete enzymes that cleave polymers, followed by microbial uptake of monomers, and production of a target product. Conversion of ferulic acid (an aromatic "monomer") is more appropriately termed "aromatic bioconversion" or something similar.*

Response: Agree. We have revised this statement. The text now reads (*see Page 2, Line 34-38*): "This study not only establishes a simple and highly efficient genome editing strategy, but also offers an encouraging example of how to apply this method to understanding microbial carbon accumulation as well as

improvement of the aromatic compound bioconversion scheme.”

7. *The authors also discuss vanillate as the “central node” for FA conversion. There are several pathways to catabolize FA that have been described. I would just remove this statement, as vanillate is certainly not the “central node” for PHA production from aromatic compounds such as ferulic acid (what does “central node” even mean?). The entire paragraph motivating FA-to-VA is quite odd and should be reworked.*

Response: Yes. We have rewritten this paragraph. The text now reads (*see Page 3, Line 65-68*):

“Moreover, vanillic acid, a intermediate metabolite in a FA catabolism pathway, has been identified as the non-fatty acid feedstock for medium-chain-length PHA (*mcl*-PHA) biosynthesis^{5, 6}, indicating the feasibility of FA-to-PHA bioconversion.”

8. *The use of the word “novel” in the scientific literature is hackneyed at this point. By definition, original research reports should be novel, so needing to claim this is redundant. This should be removed throughout the paper.*

Response: Yes. We have modified the text as suggested throughout the manuscript.

9. *Previous work has also been done on overexpression of VanAB to overcome bottlenecks in VA catabolism, including work from Yuan et al. and Salvachua et al. in *P. putida* KT2440. This work is not discussed or mentioned.*

Response: Yes. We have revised it. The text now reads (*see Page 13, Line 307-311*):

“A previous study suggested that overexpression of *vanAB* promotes VA catabolism in *P. putida* A514⁵, although their role was not reported in *P. putida* KT2440. Therefore, we inferred that increasing the expression of *vanAB*, which catabolizes VA and regenerates NAD⁺ from NADH, should counter this hurdle (Figure 3B).”

Minor concerns:

10. *Page 4, line 79. The drawbacks of specific strains are due to different engineering approaches and purposes, it’s improper to compare specific construction to one whole species here, it’s not an apple to apple comparison.*

Response: Agree. We have modified the statement. The text now reads (*see Page 4, Line 74-82*):

“However, the conversion efficiency is low, posing a challenge for the application of this organism in FA-to-PHA conversion^{7,8}. There are two major hurdles to overcome. First, FA and its intermediate compounds (e.g., vanillin) are highly toxic to the bacterial cell, inhibiting cell growth and metabolism⁴. Second, high *mcl*-PHA yields in *P. putida* are more favored to be produced from fatty acids^{9,10}. Employing non-fatty acid feedstock, especially aromatic compounds, greatly limits *mcl*-PHA accumulation^{8, 11}. Hence, metabolic engineering of *P. putida* strains is required to address these two key issues.”

11. *Page 4, line 85. The statement of here is not accurate and misleading, the disadvantages here should better be addressed respectively.*

Response: Yes. We now realize it may confuse readers and have greatly revised it. The text now reads (*see Page 4, Line 83-93*):

“A wide variety of genome-editing tools have been developed for metabolic engineering in *Pseudomonas*, the majority of which are based on homologous recombination¹². The homologous recombination mediated methods have very low mutation efficiency and are time consuming^{12, 13, 14}. Although a selection marker (e.g., *pyrF* and antibiotic resistant genes) was used to make it is easier to select for mutant, it usually leaves a marker (e.g., an antibiotic resistant gene) in the genome (Table 1)¹³, and later phage-derived recombinases (e.g., RecET and λ -Red) were employed to improve mutation efficiency^{15,16}. However, the process still leaves scars (e.g., 34 bp LoxP site) in the genome and are cumbersome for plasmid-curing (Table 1)^{1,17}. Moreover, large DNA fragment (> 5 kb) insertion has not yet reported via these homologous recombination mediated methods. .”

12. Page 4, line 89-91. The statement here is not accurate, there are many Cas9-based genome editing methods, various base editors for example. Better replace Cas9-based with Cas9-induced recombination methods.

Response: Yes. We have revised the text accordingly to ensure accurate communication. The text now reads (*see Page 4-5, Line 95-97*):

“Recently, Cas9-induced recombination methods have attracted great attention, due to their high efficiency, ease of design, short manufacturing time and low cost^{2,18}.”

13. Page 5, line 101. Replace ‘limiting their’ with ‘limiting its’.

Response: Yes. We have corrected it accordingly.

14. λ -Red or λ -red system? Please go thoroughly and correct.

Response: Yes. We have corrected it throughout the manuscript.

15. Page 21, line 507. Change to “trial-and-error”.

Response: Yes. We have revised it accordingly.

16. Page 21, line 511. Change “under gluconate” to “with gluconate as carbon source”.

Response: Yes. We have revised it.

17. Page 24, line 568. Should be ‘rpm’, no blanks.

Response: Yes. We have corrected it.

18. Page 27, line 660. ‘x’ change to ‘×’.

Response: Yes. We have corrected it accordingly.

19. Page 28, line 674 for example. There should be a blank between number and unit, please go thoroughly and correct.

Response: Yes. We have corrected it throughout the manuscript.

20. Page 31, line 745, please correct the error.

Response: Yes. We have corrected it. Please refer to “References” in the revised version.

21. Please correct all the errors in the References part, capitalize each word or the first word?

Response: Yes. We have corrected it accordingly. Please refer to “References” in the revised version.

22. Page 34, line 825, *phaZ* should be italic.

Response: I referred the literature ¹⁹ and noticed that the “PhaZ” was used to represent the PhaZ depolymerase protein. Therefore, we have decided to keep it. Please refer to **Page 40, Line 937-940** in the revised version.

23. Supplementary Page 1, line 5, please correct the error.

Response: Yes. We have corrected it accordingly. Please refer to **Page 1, Line 3-8** in the revised supplementary materials.

Reviewer #2 (Remarks to the Author):

This manuscript describes the development of a highly efficient CRISPR/Cas9n genome editing strategy (CRP) in *Pseudomonas putida*. Also reported is the application of CRP for improving FA-to-PHA bioconversions. The experiment data is substantial and analyzed deeply.

Major

24. L77-78 mentions ‘avoiding the drawbacks (e.g., inefficient heterologous gene expression and plasmid instabilities) of recombinant strains, such as *Escherichia coli*’. The description of drawbacks for the recombinant *Escherichia coli* is not persuasive.

Response: Yes. We have realized the statement was not accurate and revised it. Please refer to **Page 4, Line 74-82** in the revised version.

25. L115 mentions the expression of nine genes that are divided into four engineered modules. However, Fig 5(A) showed eight genes were integrated into genome and two genes were deleted.

Response: Six different genes were integrated into the KTc9n genome, including *ech*, *vdh*, *fcs*, *vanA*, *vanB* and mutated *phaC1*. Among them, two copies of *vanAB* were inserted into the genome, while others were a single copy. In addition, three genes, *ferR*, *tesB* and *tesBII*, were deleted. Together, it’s nine different genes that were modified. We have revised it to ensure accurate communication.

The text now reads (*see* **Page 6, Line 122-132 and Figure 5A**):

“Via this developed genome editing tool, we successfully engineered *P. putida* KT2440 to enhance FA-to-PHA conversion by modulating the expression of nine different genes that are divided into four engineered modules. The enzymes and regulator of Module 1 (*ech*, *vdh*, *fcs* and *ferR*) together degrade FA and generate vanillic acid, and Module 2 (*vanAB* genes) converts vanillic acid (VA) to protocatechuic acid (PCA) to alleviate VA accumulation and redox imbalance. Module 3 includes the deletion of two thioesterase genes (*tesB* and *tesBII*) enabling the metabolic flux from the fatty acid biosynthesis pathway to *mcl*-PHA biosynthesis, while Module 4 (*phaC1*) synthesizes *mcl*-PHA from the precursor, 3-hydroxyacyl-CoA. Each module is incorporated into the KT2440 genome.”

26. L124 In this study, more than 20 strains were constructed to improve *mcl*-PHA

production. Which strains would produce 270 mg/L *mcl*-PHA? This section needs more description.

Response: Yes. We have revised the text to improve clarity. The text now reads (*see* Page 6, Line 132-135):

“Through metabolic engineering and optimization of cultivation conditions, we constructed the KTc9n20 strain to improve the *mcl*-PHA production from consumed FA substrate (20 mM) to ~270 mg/L, laying the foundation for a FA bioprocessing platform.”

27. L131 This study first tested the reported two-plasmid CRISPR/Cas9 system and a CRISPR/Cas9- λ -red system in *P. putida* KT2440, respectively. This section is vague and difficult to understand. If you had tested this reported system, please add more description.

Response: Yes. We have rewritten this paragraph to ensure clarity.

The text now reads (*see* Page 6-8, Line 139-175):

“Previously established CRISPR/Cas9 systems in *P. putida* KT2440 are divided into two-plasmid CRISPR/Cas9- λ -Red and three-plasmid CRISPR/Cas9-Ssr systems^{1,20}. This study first tested the CRISPR/Cas9- λ -Red system in *P. putida* KT2440 due to its high mutation efficiency. Meanwhile the CRISPR/Cas9 system was used as the negative control¹. The *pyrF* gene (PP_1815), encoding orotidine-5'-phosphate decarboxylase (ODCase), was chosen as the target, as deletion of this gene would generate uracil auxotrophic and 5-fluoroorotic acid (5-FOA) resistant phenotypes²¹.

Highly variable mutation efficiency of a CRISPR/Cas9- λ -Red system. To investigate the system, the pCas9 plasmid was first constructed, which expresses Cas9 protein under the control of the constitutive P_{min} promoter²². Subsequently, the λ -Red expression cassette (P_{xylA}::*gam-bet-exo*) was introduced into pCas9 to improve repairing efficiency and thus alleviate the toxicity of Cas9 caused DSB. It generated the construct pCas9-Red (Table 2). Meanwhile, the other vector, pBBR1-1²³, was constructed to express the customized sgRNA for *pyrF* under control of the P_{trc} promoter and contain 0.5 kb homologous repairing arms, providing homology-directed repair to fix DNA lesions (Table 2 and Supplementary S1). The pCas9 and pBBR1-1 were co-transformed into KT2440. Consistent with previous reports^{1,21,24}, the co-expression vectors produced cells with neither antibiotic resistance nor 5-FOA resistance, whereas two controls (KT2440 carrying either pCas9 or pBBR1-1) generated antibiotic-resistant transformants (Table 3). Together, this suggested Cas9-induced chromosomal cleavage is indeed toxic in *P. putida* KT2440, at least at the selected gene locus. In contrast, when pCas9-Red and pBBR1-1 were co-transformed into KT2440, dozens of colonies were observed on 5-FOA+Gm^r+Km^r LB selection plates. This indicated *pyrF* was knocked-out, and was further confirmed by colony PCR screening and DNA sequencing. Hence, this CRISPR/Cas9- λ -Red system works in KT2440, consistent with the previous study¹.

However, the calculated mutation efficiency for *pyrF* was 1/7730 (Table 3), far from previously reported high mutation efficiency (70-100%) via the CRISPR/Cas9- λ -Red system¹. We speculated that the low mutation rate for *pyrF* was likely due to variant efficiency among various genes³³. Moreover, considering that various vectors with different copy numbers were used to

express the four components (Cas9, the gRNA cassette, λ -Red system and the homologous repairing arms) between this study and previous work ¹, expression dose of the four functional elements was also a factor contributing to the varied mutation rates (Table 3 and Table S2). Therefore, highly variable editing efficiency of CRISPR/Cas9- λ -Red system restricts whole genome editing in bacteria in general, and certain genes of *P. putida* in particular, which have either low repairing ability or poor mutation rate.”

In addition, we have inserted methodology description about these two systems in the “Supplementary materials S1” (see Page 2-3, Line 50-66 in the revised Supplementary materials).

28. L228 Four different length genes or gene clusters were integrated into the chromosome of KTc9n. Which gene loci were used for the integration of these different length genes or gene cluster? How many colonies were picked from the plates for identification in each gene integration experiment? These details are very important for a highly efficient genome editing system.

Response: Yes. We have rewritten this paragraph and inserted a significant amount of detailed descriptions (Table S3) to ensure accurate communication and it now reads (see Page 10-11, Line 241-255 and Table S3):

“Application of CRP genome editing system in *P. putida* KT2440. In addition to the *icd* gene, the mutant efficiencies for other locations in the KT2440 strain were systematically investigated (Table S3). The deletion efficiencies were tested for in either single-gene/gene cluster or double-gene with different deleted lengths (from 98 bp to 4643 bp) and were found to be 100% (Table S3). Meanwhile, the efficiency for gene integration into *P. putida* chromosome was also examined. DNA fragments with different lengths (from 1900 bp to 15000 bp) were successfully integrated into the chromosome of KTc9n, respectively (Table S3). Furthermore, the efficiency of simultaneous gene insertion and deletion in the KTc9n genome (KTc9n13) also achieved 100%. For each mutant, 8-20 colonies were randomly picked from the selection plates (LB+5FOA+Ura⁺) for PCR and DNA sequencing, and found that all contained the desired mutation (Table S3). Hence, our results suggested that the mutation efficiency of this method theoretically can reach 100%, which has an advantage in genome editing events with low efficiency, such as a large DNA fragment (> 5 kb) editing.”

29. L285 In addition, compared to KTc9n, the cell growth of KTc9n4 was also improved, which should be the result of more utilized VA, as a carbon source, to support the growth KTc9n4, in addition to FA. Why would vanillic acid(VA) be used as a carbon source? Please add the relating reference.

Response: We now realize the original statement confused readers. In this revised version, we have re-written this part, both to improve the readability and to present the new experiments to support our result. Please refer to Page 12-13, Line 294-345.

30. L433 variable genome editing events, including single-gene deletion, double-gene disruption, and large DNA fragment integration (>10 kb), respectively, were tested in this system. I can't find the double-gene disruption data in this paper. Please provide more accurate information.

Response: Yes. Please refer to our response to Review-Comment No. 28 above.

In addition, we have inserted methodology description about double-gene disruption

in the “Materials and Methods”. The text now reads (*see Page 27, Line 663-667 and Table S3*):

“For *tesB* and *tesBII* double-gene disruption, pBBR1-7 was transformed into KTc9n to delete *tesB*, while pBBR1-8 was subsequently introduced to remove *pyrF* marker, generating the mutant KTc9n11. In the following round of genome editing, pBBR1-19 and pBBR1-20 were sequentially transformed into KTc9n11 to knockout *tesBII*. Consequently, KTc9n15 was constructed.”

31. L624 ‘each of the plasmids (1 µg) were transformed into KT2440 by electroporation’ should be ‘each of the plasmids (1 µg) was transformed into KT2440 by electroporation’.

Response: Yes. We have corrected the text accordingly.

Reviewer #3 (Remarks to the Author):

Authors develop a genome editing strategy (CRP) for *P. putida* KT2440 using an integrated CRISPR/Cas9n-λ red system with *pyrF* as a selection marker, which maintains cell viability and strain genetic stability, increases mutation efficiency, and simplifies manipulation. Four functional modules consisting of nine genes involved in FA catabolism and polyhydroxyalkanoate (PHA) biosynthesis, were integrated into the KT2440 genome to improve FA-to-PHA bioconversion. The study establishes a simple and highly efficient genome editing strategy. A few issues need be addressed:

32. *P. putida* KT2440 can already produce many grams PHA. This study showed only mg level PHA. Something must be wrong with the growth and PHA production;

Response: That’s an excellent point. The PHA production in *P. putida* KT2440 greatly depends on the types of carbon substrates. High *mcl*-PHA titer (e.g., 26 g/L) in *P. putida* are more favored to be produced from fatty acids^{9, 10}. Employing non-fatty acid feedstock, especially aromatic compounds, greatly limits *mcl*-PHA accumulation^{8, 11} (*see Page 4, Line 78-82 in the revised version*). The Guss group, as mentioned by Reviewer 1, also reported mg level PHA was produced by *P. putida* KT2440 when either lignin-containing stream or aromatic compounds was used as the carbon substrate⁸.

In addition, the different measurement methods of PHA production also affect the result. It is more accurate when a GC or GC-MS method was used. In contrast, the measured PHA production is commonly higher than the actual value when either the weighing method or Nile red staining was used. We have summarized the detailed information about *mcl*-PHA production in *Pseudomonas* strains from aromatic compounds in Table 4. Please refer to Table 4 in the revised version.

33. What kind of PHA was produced, *scl*- or *mcl* PHA? What *mcl* PHA was produced? Not show at all.

Response: The produced PHA were medium-chain-length PHAs. We have stated this throughout the manuscript and provided the detailed information in Table S4.

The text now reads (*see Page 17, Line 404-407 and Table S4*):

“The accumulated *mcl*-PHA in KTc9n15 was slightly higher than that of single knockout mutant, KTc9n12, increased to 164 mg/L, while KTc9n16 did not further gains in *mcl*-PHA accumulation (105 mg/L), compared to KTc9n15

(Figure 4C and Table S4).”

Another example of our revision is (*see Page 23, Line 564-567*):

“Considering that accumulated *mcl*-PHAs have a heterogeneous monomer composition, with 6-12 carbon atoms (Table S4), such different effects are probably due to the different substrate specificities of the thioesterases.”

References cited in this response letter:

1. Sun J, *et al.* Genome editing and transcriptional repression in *Pseudomonas putida* KT2440 via the type II CRISPR system. *Microb Cell Fact* **17**, 018-0887 (2018).
2. Jiang W, Bikard D, Cox D, Zhang F, Marraffini LA. RNA-guided editing of bacterial genomes using CRISPR-Cas systems. *Nat Biotechnol* **31**, 233-239 (2013).
3. Blatny JM, Brautaset T, Winther-Larsen HC, Haugan K, Valla S. Construction and use of a versatile set of broad-host-range cloning and expression vectors based on the RK2 replicon. *Appl Environ Microbiol* **63**, 370-379 (1997).
4. Graf N, Altenbuchner J. Genetic engineering of *Pseudomonas putida* KT2440 for rapid and high-yield production of vanillin from ferulic acid. *Appl Microbiol Biotechnol* **98**, 137-149 (2014).
5. Lin L, *et al.* Systems biology-guided biodesign of consolidated lignin conversion. *Green Chem* **18**, 5536-5547 (2016).
6. Wang X, *et al.* Simultaneous improvements of *Pseudomonas* cell growth and polyhydroxyalkanoate production from a lignin derivative for lignin-consolidated bioprocessing. *Appl Environ Microbiol* **84**, 01469-01418 (2018).
7. Linger JG, *et al.* Lignin valorization through integrated biological funneling and chemical catalysis. *Proc Natl Acad Sci USA* **111**, 12013-12018 (2014).
8. Salvachua D, *et al.* Metabolic engineering of *Pseudomonas putida* for increased polyhydroxyalkanoate production from lignin. *Microb Biotechnol* **30**, 1751-7915 (2019).
9. Gao J, Vo MT, Ramsay JA, Ramsay BA. Overproduction of MCL-PHA with high 3-hydroxydecanoate Content. *Biotechnol Bioeng* **115**, 390-400 (2018).
10. Chung A, Liu Q, Ouyang SP, Wu Q, Chen GQ. Microbial production of

- 3-hydroxydodecanoic acid by *pha* operon and *fadBA* knockout mutant of *Pseudomonas putida* KT2442 harboring *tesB* gene. *Appl Microbiol Biotechnol* **83**, 513-519 (2009).
11. Meng DC, Chen GQ. Synthetic biology of polyhydroxyalkanoates (PHA). *Adv Biochem Eng Biotechnol* **162**, 147-174 (2018).
 12. Martinez-Garcia E, de Lorenzo V. Engineering multiple genomic deletions in gram-negative bacteria: analysis of the multi-resistant antibiotic profile of *Pseudomonas putida* KT2440. *Environ Microbiol* **13**, 2702-2716 (2011).
 13. Galvao TC, de Lorenzo V. Adaptation of the yeast URA3 selection system to gram-negative bacteria and generation of a $\Delta betCDE$ *Pseudomonas putida* strain. *Appl Environ Microbiol* **71**, 883-892 (2005).
 14. Graf N, Altenbuchner J. Development of a method for markerless gene deletion in *Pseudomonas putida*. *Appl Environ Microbiol* **77**, 5549-5552 (2011).
 15. Luo X, Yang Y, Ling W, Zhuang H, Li Q, Shang G. *Pseudomonas putida* KT2440 markerless gene deletion using a combination of λ Red recombineering and Cre/loxP site-specific recombination. *FEMS Microbiol Lett* **363**, 21 (2016).
 16. Wenzel SC, Gross F, Zhang Y, Fu J, Stewart AF, Muller R. Heterologous expression of a myxobacterial natural products assembly line in *Pseudomonads* via Red/ET recombineering. *Chem Biol* **12**, 349-356 (2005).
 17. Hoang TT, Karkhoff-Schweizer RR, Kutchma AJ, Schweizer HP. A broad-host-range Flp-FRT recombination system for site-specific excision of chromosomally-located DNA sequences: application for isolation of unmarked *Pseudomonas aeruginosa* mutants. *Gene* **212**, 77-86 (1998).
 18. Ran FA, *et al.* Double nicking by RNA-guided CRISPR Cas9 for enhanced genome editing specificity. *Cell* **154**, 1380-1389 (2013).
 19. de Eugenio LI, *et al.* The turnover of medium-chain-length polyhydroxyalkanoates in *Pseudomonas putida* KT2442 and the fundamental role of PhaZ depolymerase for the metabolic balance. *Environ Microbiol* **12**, 207-221 (2010).
 20. Aparicio T, de Lorenzo V, Martinez-Garcia E. CRISPR/Cas9-based counterselection boosts recombineering efficiency in *Pseudomonas putida*. *Biotechnol J* **13**, 4 (2018).

21. Xu T, *et al.* Efficient genome editing in *Clostridium cellulolyticum* via CRISPR-Cas9 nickase. *Appl Environ Microbiol* **81**, 4423-4431 (2015).
22. Miller WG, Leveau JH, Lindow SE. Improved *gfp* and *inaZ* broad-host-range promoter-probe vectors. *Mol Plant Microbe Interact* **13**, 1243-1250 (2000).
23. Hao L, Liu X, Wang H, Lin J, Pang X. Detection and validation of a small broad-host-range plasmid pBBR1MCS-2 for use in genetic manipulation of the extremely acidophilic *Acidithiobacillus* sp. *J Microbiol Methods* **90**, 309-314 (2012).
24. Jiang Y, *et al.* CRISPR-Cpf1 assisted genome editing of *Corynebacterium glutamicum*. *Nat Commun* **8**, (2017).

In conclusion, we greatly appreciate to the reviewers' very helpful comments, and hope that these changes and responses now adequately address the concerns raised. All changes have been tracked in the revised manuscript.

REVIEWERS' COMMENTS:

Reviewer #1 (Remarks to the Author):

The authors have revised the manuscript substantially and included new data. There are still several items that need to be addressed. The Discussion section makes claims about the PHA titers that are, I believe, not correct and this should be carefully considered. Regardless, having the "highest titer" reported is not really that much to boast about when the experiments are still done in shake flasks and not bioreactors in a fed-batch mode...essentially the paper is still strong and of high quality without having to say that this is the "best", especially when it likely is not the highest titer from aromatic compounds in this background strain.

Related to the first sentence in the Introduction, there is a debate as to whether lignin or chitin is the second most abundant carbon compound on earth, so either please provide an *original* citation showing this (not just another paper that says this with no citation) or modify this statement.

Ferulic acid is linked to hemicellulose and lignin, and calling it a "component" of lignin is a bit misleading in a strict sense of the definition of lignin being made up of monolignols. The authors may wish to rethink the wording there.

The connection of FA to lignin is a bit tenuous and the papers cited (1, 3) are not really about microbial lignin valorization. Would suggest that the authors cited reviews and seminal papers in the field of microbial lignin valorization here.

Reference 7 has nothing to do with PHAs, so not sure why that is cited there. Rather that is a perspective/review about why lignin valorization is important.

Linger et al. PNAS (ref 14) and Salvachua et al. Green Chem (mentioned in the previous review) both produce PHAs from FA and VA as well. Not sure why these are not included in statements regarding FA and VA conversion to PHAs.

Figure 3D could use some editing. I would suggest that the authors more clearly delineate the 3 bars from each other. Perhaps make the bars thinner to separate the 3-bar sets from each other more. That would be helpful for visual clarity.

I noted in my previous review that work has been done in *P. putida* KT2440 as well to show that VA accumulates from FA. This was reported in Salvachua et al. Green Chem 2018, which the authors ignored in the edits. Here is the link to the paper: <https://pubs.rsc.org/en/content/articlehtml/2018/gc/c8gc02519c>.

phaZ knockout and phaC1 over-expression have been done in Salvachua et al. Microbial Biotech 2019 as well, so where the authors note "it was speculated that..." on line 390 is

known from previous literature. This previous paper is highly relevant to this work, but is not included in the Discussion, nor is the titer reported therein compared to this study (where the authors note that this is the highest PHA production from aromatic compounds in *P. putida*). Is that actually the case? Ditto for the work from Linger et al. PNAS 2014 and multiple works from Joshua Yuan's group?

After these (fairly minor) changes are made, I think that this paper is ready to publish. The genetic engineering methodology outlined here will find utility and the metabolic engineering results will be useful to the many groups working on PHA production in Pseudomonads.

Reviewer #2 (Remarks to the Author):

The authors answered and replied to all my comments properly.
Thank you for the time and effort expended in improving your work!

Reviewer #3 (Remarks to the Author):

The authors have addressed my concerns, the paper can be accepted.

Reviewers' comments:

Reviewer #1 (Remarks to the Author):

1. *The authors have revised the manuscript substantially and included new data. There are still several items that need to be addressed. The Discussion section makes claims about the PHA titers that are, I believe, not correct and this should be carefully considered. Regardless, having the "highest titer" reported is not really that much to boast about when the experiments are still done in shake flasks and not bioreactors in a fed-batch mode...essentially the paper is still strong and of high quality without having to say that this is the "best", especially when it likely is not the highest titer from aromatic compounds in this background strain.*

Response: Yes. We have modified this statement. The text now reads (*see Page 17-18, Line 419-423*):

“Our study divided the nine genes known to be involved in the pathways into four modules, which facilitated the characterization of each aspect of the process, and consequently developed the engineered KTe9n20 strain with greatly improved ferulic acid-to-*mcl*-PHA conversion (Table 4).”

2. *Related to the first sentence in the Introduction, there is a debate as to whether lignin or chitin is the second most abundant carbon compound on earth, so either please provide an *original* citation showing this (not just another paper that says this with no citation) or modify this statement.*

Response: Yes. We have modified this statement and cited the literature. The text now reads (*see Page 3, Line 51-54*):

“Considering that lignocellulose is an abundant carbon compound on the earth¹, investigation of ferulic acid catabolism not only provides insights into the global carbon cycle², but also contributes to the development in biocatalytic conversion of ferulic acid to valuable bioproducts^{3,4}.”

3. *Ferulic acid is linked to hemicellulose and lignin, and calling it a “component” of lignin is a bit misleading in a strict sense of the definition of lignin being made up of monolignols. The authors may wish to rethink the wording there.*

Response: Yes. We have revised it. The text now reads (*see Page 3, Line 48-51*):

“Ferulic acid is a ubiquitous phenolic component of lignocellulose, which is linked to hemicellulose and lignin via ester linkages, and is released to the environment by fungal and bacterial esterase enzymes^{5,6,7}. Moreover, ferulic acid is an intermediate metabolite during lignin biodegradation^{8,9}.”

4. *The connection of FA to lignin is a bit tenuous and the papers cited (1, 3) are not really about microbial lignin valorization. Would suggest that the authors cited reviews and seminal papers in the field of microbial lignin valorization here.*

Response: Yes. We have revised it accordingly. Please refer to our response to Review-Comment No. 2 above.

5. *Reference 7 has nothing to do with PHAs, so not sure why that is cited there. Rather that is a perspective/review about why lignin valorization is important.*

Response: Yes. We have revised it. The text now reads (*see Page 3, Line 57-59*):

“Recently there has been strong interest in the use of bacteria to produce more diverse and valuable products (e.g., polyhydroxyalkanoates (PHAs) and muconic acid) from ferulic acid^{4,10,11}.”

6. *Linger et al. PNAS (ref 14) and Salvachua et al. Green Chem (mentioned in the previous review) both produce PHAs from FA and VA as well. Not sure why these are not included in statements regarding FA and VA conversion to PHAs.*

Response: That’s an excellent suggestion! I have cited these two papers to support our statement, although *Salvachua et al. Green Chem* produce muconic acid (MA) from FA. Please refer to our response to Review-Comment No. 5 above.

7. *Figure 3D could use some editing. I would suggest that the authors more clearly delineate the 3 bars from each other. Perhaps make the bars thinner to separate the 3-bar sets from each other more. That would be helpful for visual clarity.*

Response: Yes. We have modified it. Please refer to Figure 3d in the revised version.

8. *I noted in my previous review that work has been done in *P. putida* KT2440 as well to show that VA accumulates from FA. This was reported in *Salvachua et al. Green Chem 2018*, which the authors ignored in the edits. Here is the link to the paper: <https://pubs.rsc.org/en/content/articlehtml/2018/gc/c8gc02519c>.*

Response: Thanks for the helpful suggestion! We have cited it and revised the text as suggested. The text now reads (*see Page 11, Line 254-258*):

“Previous studies suggested that overexpression of *vanAB* promotes vanillic acid catabolism in *P. putida* KT2440¹¹ and A514¹², respectively. Therefore, we inferred that increasing the expression of *vanAB*, which regenerates NAD⁺ from NADH, should counter this hurdle (Figure 3b).”

9. *phaZ knockout and phaC1 over-expression have been done in *Salvachua et al. Microbial Biotech 2019* as well, so where the authors note “it was speculated that...” on line 390 is known from previous literature. This previous paper is highly relevant to this work, but is not included in the Discussion, nor is the titer reported therein compared to this study (where the authors note that this is the highest PHA production from aromatic compounds in *P. putida*). Is that actually the case? Ditto for the work from *Linger et al. PNAS 2014* and multiple works from Joshua Yuan’s group?*

Response: Yes. We have revised it. The text now reads (*see Page 13, Line 312-315*):

“In addition, it’s known that *phaZ* (*mcl*-PHA depolymerase) knockout and *phaC* (*mcl*-PHA synthase) knockin enhance *mcl*-PHA accumulation in *P. putida*¹³. Therefore, *phaZ* was deleted and replaced by a copy of *phaC1*, generating the mutant KTe9n13 (Table 1 and Supplementary Table 3).”

In addition, we have summarized the *Pseudomonas mcl*-PHA production from aromatic compounds in Table 4 according to the highly relevant works. It’s included in Results and Discussion. One example of our revision is (*see Page 18, Line 433-435 and Table 4*):

“However, the effects on *mcl*-PHA accumulation are variable under different conditions, such as different feedstock. Thereby, these strategies must be examined in a trial-and-error type of approach (Table 4).”

References cited in this response letter:

1. Ragauskas AJ, *et al.* The path forward for biofuels and biomaterials. *Science* **311**, 484-489 (2006).
2. Jiao N, Tang K, Cai H, Mao Y. Increasing the microbial carbon sink in the sea by reducing chemical fertilization on the land. *Nat Rev Microbiol* **9**, 75-75 (2011).
3. Graf N, Altenbuchner J. Genetic engineering of *Pseudomonas putida* KT2440 for rapid and high-yield production of vanillin from ferulic acid. *Appl Microbiol Biotechnol* **98**, 137-149 (2014).
4. Linger JG, *et al.* Lignin valorization through integrated biological funneling and chemical catalysis. *Proc Natl Acad Sci USA* **111**, 12013-12018 (2014).
5. Bugg TD, Ahmad M, Hardiman EM, Rahmanpour R. Pathways for degradation of lignin in bacteria and fungi. *Nat Prod Rep* **28**, 1883-1896 (2011).
6. Chateigner-Boutin AL, *et al.* Ferulate and lignin cross-links increase in cell walls of wheat grain outer layers during late development. *Plant Sci* **276**, 199-207 (2018).
7. Zhang Y, *et al.* *Bacillus methylotrophicus* isolated from the cucumber rhizosphere degrades ferulic acid in soil and affects antioxidant and rhizosphere enzyme activities. *Plant Soil* **392**, 309-321 (2015).
8. Anderson EM, *et al.* Differences in S/G ratio in natural poplar variants do not predict catalytic depolymerization monomer yields. *Nat Commun* **10**, 019-09986 (2019).
9. Lin L, Wang X, Cao L, Xu M. Lignin catabolic pathways reveal unique characteristics of dye-decolorizing peroxidases in *Pseudomonas putida*. *Environ Microbiol* **21** (5), 1847-1863 (2019).
10. Ni J, Gao YY, Tao F, Liu HY, Xu P. Temperature-directed biocatalysis for the sustainable production of aromatic aldehydes or alcohols. *Angew Chem Int Ed Engl* **57**, 1214-1217 (2018).
11. Salvachúa D, *et al.* Bioprocess development for muconic acid production from aromatic compounds and lignin. *Green Chemistry* **20**, 5007-5019 (2018).

12. Lin L, *et al.* Systems biology-guided biodesign of consolidated lignin conversion. *Green Chem* **18**, 5536-5547 (2016).
13. Salvachua D, *et al.* Metabolic engineering of *Pseudomonas putida* for increased polyhydroxyalkanoate production from lignin. *Microb Biotechnol* **30**, 1751-7915 (2019).

In conclusion, we greatly appreciate to the reviewers' very helpful comments, and hope that these changes and responses now adequately address the concerns raised. All changes have been tracked in the revised manuscript.